# Accelerating Large Language Model Reasoning via Speculative Search

**Zhihai Wang** [1]   **Jie Wang** [1✉]   **Jilai Pan** [1]   **Xilin Xia** [1]   **Huiling Zhen** [2]   **Mingxuan Yuan** [2]   **Jianye Hao** [2 3]   **Feng Wu** [1]

## Abstract

Tree-search-based reasoning methods have significantly enhanced the reasoning capability of large language models (LLMs) by facilitating the exploration of multiple intermediate reasoning steps, i.e., thoughts. However, these methods suffer from substantial inference latency, as they have to generate numerous reasoning thoughts, severely limiting LLM applicability. To address this challenge, we propose a novel Speculative Search (SpecSearch) framework that significantly accelerates LLM reasoning by optimizing thought generation. Specifically, SpecSearch utilizes a small model to strategically collaborate with a large model at both thought and token levels, efficiently generating high-quality reasoning thoughts. The major pillar of SpecSearch is a novel quality-preserving rejection mechanism, which effectively filters out thoughts whose quality falls below that of the large model's outputs. Moreover, we show that SpecSearch preserves comparable reasoning quality to the large model. Experiments on both the Qwen and Llama models demonstrate that SpecSearch significantly outperforms state-of-the-art approaches, achieving up to 2.12× speedup with comparable reasoning quality. Code is available at https://github.com/MIRALab-USTC/LLMReasoning-SpecSearch.

## 1. Introduction

The reasoning capabilities of large language models (LLMs) have significantly advanced with the adoption of slow-thinking processes based on tree-search-based (TSB) rea-

This work was done when Zhihai Wang was an intern at Huawei. [1]MoE Key Laboratory of Brain-inspired Intelligent Perception and Cognition, University of Science and Technology of China [2]Noah's Ark Lab, Huawei Technologies [3]College of Intelligence and Computing, Tianjin University. Correspondence to: Jie Wang <jiewangx@ustc.edu.cn>.

*Proceedings of the 42$^{nd}$ International Conference on Machine Learning*, Vancouver, Canada. PMLR 267, 2025. Copyright 2025 by the author(s).

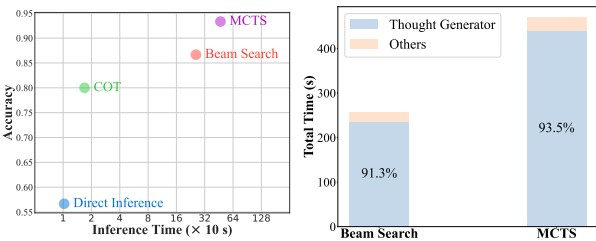

| (a) Latency-Accuracy | (b) Bottleneck of Reasoning |
|---|---|

*Figure 1.* (a) The inference latency increases by several orders of magnitude with the introduction of tree-search-based reasoning methods. (b) Thought generation acts as an efficiency bottleneck of tree-search-based reasoning methods.

soning methods (Yao et al., 2023; Wan et al., 2024a; Jiang et al., 2024; Wu et al., 2024). These TSB methods enhance reasoning by following the Chain-of-Thought (COT) approach (Wei et al., 2022), which decomposes problem-solving into a sequence of intermediate reasoning steps, termed thoughts. Building upon this, TSB frameworks such as Tree-of-Thoughts (TOT) (Yao et al., 2023) integrate thought generation and evaluation with search algorithms—such as beam search (Kang et al., 2024) and Monte Carlo Tree Search (MCTS) (Chen et al., 2024; Zhang et al., 2024b)—to systematically explore diverse reasoning paths. By incorporating these techniques, TSB methods provide LLMs with a deliberate and structured reasoning framework, significantly improving their capability to tackle complex and multi-step reasoning tasks.

However, existing TSB reasoning methods often suffer from substantial inference latency (Gao et al., 2024; Wang et al., 2024e), with inference latency increasing by several orders of magnitude (see Figure 1a). The primary bottleneck stems from the need to explore a vast number of reasoning thoughts (see Figure 1b). This substantial increase in inference latency poses significant challenges for practical deployment, particularly in real-time applications requiring low-latency performance (Zhou et al., 2024; Xia et al., 2024). However, effective and efficient strategies to accelerate slow-thinking reasoning in LLMs without compromising reasoning quality remain largely underexplored.

In this paper, we propose Speculative Search (SpecSearch), a novel and efficient framework that significantly accelerates LLM reasoning while maintaining comparable quality. At its core, SpecSearch features a bi-level speculative

thought generator, where a small model strategically collaborates with a large model at both coarse-grained thought and fine-grained token levels. This innovative design optimizes thought generation efficiency, enabling faster yet effective reasoning. To ensure reasoning quality, SpecSearch proposes to filter out thoughts that fall below the quality of the large model's outputs. SpecSearch achieves this by accurately and efficiently estimating the quality through a non-parametric statistical estimation method, leveraging historical reasoning thoughts from the large model. Moreover, we establish a theoretical guarantee that SpecSearch preserves comparable reasoning quality to the large model.

To demonstrate the effectiveness of SpecSearch, we evaluate it on two complex reasoning datasets: MATH and GSM8K. Experiments using both the Qwen and Llama models demonstrate that our method significantly outperforms state-of-the-art (SOTA) approaches, achieving up to 2.12× speedup while maintaining comparable reasoning quality. Moreover, experiments demonstrate that SpecSearch seamlessly integrates with several tree search algorithms and thought evaluators, delivering substantial acceleration without compromising reasoning quality. These results underscore SpecSearch's ability to significantly enhance the efficiency of existing TSB reasoning methods.

We summarize our major contributions as follows. (1) **A Novel SpecSearch Framework** Observing that thought generation is a major efficiency bottleneck, we propose SpecSearch, which utilizes a small model collaborating with a large model at both coarse-grained thought and fine-grained token levels. This design significantly improves thought generation efficiency, thereby accelerating LLM reasoning. (2) **Quality-Preserving Rejection Mechanism** To ensure high reasoning quality, we propose to filter out thoughts whose quality falls below that of the large model's outputs, and efficiently estimate the large model's quality via its historical reasoning thoughts. (3) **Theoretical Guarantee** We provide a theoretical analysis showing that SpecSearch preserves reasoning quality comparable to that of the large model. (4) **Significant Speedup and Versatility** Experiments demonstrate that SpecSearch significantly outperforms SOTA approaches, achieving up to 2.12× speedup while preserving comparable reasoning quality. Moreover, experiments demonstrate the strong compatibility of SpecSearch with different LLMs, search algorithms, and thought evaluators, highlighting its broad applicability.

## 2. Related Work

**Speculative Decoding** As the number of parameters in LLMs increases, inference latency has become a major obstacle to their broader applications (Zhou et al., 2024; Wan et al., 2024b; Xia et al., 2024; Zhang et al., 2024a). This latency is primarily caused by the autoregressive decoding process, where each token is generated sequentially, depen-

dent on the preceding token's completion (Lu et al., 2024; Xia et al., 2024). To accelerate LLM decoding, an innovative paradigm (Leviathan et al., 2023; Chen et al., 2023a;b; Yang et al., 2024; Li et al., 2024; Kou et al., 2024; Zhong & Bharadwaj, 2024) introduces the idea of speculative execution (Burton, 1985; Hennessy & Patterson, 2012) as in computer architecture to LLM decoding in a draft-then-verify style. Specifically, speculative decoding methods speculatively draft a sequence of tokens via a small model, and then verify the sequence via a large model in parallel, thus speeding up the LLM decoding process (see Figure 6a in Appendix B). However, speculative decoding—a token-level inference acceleration method—can be poorly aligned with optimizing search-based reasoning approaches that involve intricate, non-sequential reasoning thought generation, leading to suboptimal acceleration performance. Inspired by speculative execution, we propose a novel SpecSearch framework to leverage the inherent structure of TSB reasoning frameworks by formulating both thought and token generation as speculative tasks. To the best of our knowledge, we are *the first* to well generalize speculative execution to TSB reasoning, providing a novel speculative execution formulation for accelerating LLM reasoning. Moreover, we provide a detailed discussion on novelty of SpecSearch over standard speculative decoding and TreeBon (Qiu et al., 2024) in Appendix G.

**The Novelty of SpecSearch over Speculative Decoding Methods** We discuss the novelty of SpecSearch compared to existing SD techniques, emphasizing key distinctions in terms of **speculative formulation**, **verification** and **rejection strategies**, and **theoretical guarantees**. (1) **Bi-Level Speculative Formulation**: Unlike existing SD methods focused solely on tokens, SpecSearch treats both high-level thoughts and low-level tokens as bi-level speculative tasks. This enables 1) **Structural Alignment** with reasoning frameworks, where thoughts are fundamental units, and 2) **Compatibility** with standard SD methods through low-level token-level speculation. (2) **Contextual Verification for Higher Acceptance and Speedup**: Unlike existing SD methods that enforce strict token-level alignment, leading to frequent rejections, SpecSearch verifies the **contextual quality** of reasoning thoughts. This allows acceptance of correct but non-aligned outputs, substantially boosting acceptance rates and achieving significant speedups. (3) **Quality-Preserving Rejection Mechanism**: Unlike token-level rejection in standard SD methods, SpecSearch proposes **quality-preserving thought-level rejection** based on contextual quality. It discards entire thoughts only when their quality is lower than the large model's, ensuring high-quality reasoning throughout decoding. (4) **Theoretical Guarantee of Reasoning Quality**: While standard SD methods preserve token-level distributions, SpecSearch guarantees that reasoning quality remains comparable with

outputs from the large model.

**Tree-Search-Based Reasoning Acceleration** In recent years, tree-search-based reasoning methods (Yao et al., 2023; Hao et al., 2023; Hui et al., 2024; Wan et al., 2024a; Jiang et al., 2024; Wu et al., 2024; Xie et al., 2023) have significantly enhanced the reasoning capabilities of LLMs. To accelerate tree-search-based reasoning, Gao et al. (2024) directly integrate standard speculative decoding techniques with reasoning methods. Subsequently, SEED (Wang et al., 2024e) proposes a Scheduled Speculative Decoding method, which manages the scheduling of parallel small models based on only one shared large model. These methods improve the efficiency of tree-search-based reasoning methods, demonstrating the potential of designing speculative execution strategies in the LLM reasoning framework. However, these methods primarily design speculative execution strategies at the token level, neglecting the inherent structure of LLM frameworks, where reasoning thoughts are fundamental units. This oversight results in suboptimal acceleration performance. In contrast, our SpecSearch proposes a novel bi-level speculative thought generator, which utilizes a small model collaborating with a large model at both coarse-grained thought and fine-grained token levels, leading to significant acceleration with comparable quality.

**The Novelty of SpecSearch over TreeBon (Qiu et al., 2024)** We discuss the novelty of SpecSearch compared to Treebon (Qiu et al., 2024), emphasizing key distinctions in terms of **motivation**, **speculative formulation**, **rejection strategies**, and **theoretical guarantees**. (1) **Distinct Motivation**: Unlike Treebon, which targets accelerating best-of-n sampling through speculative rejection combined with tree search, SpecSearch is the first to well generalize speculative execution to tree-based LLM reasoning. (2) **Bi-Level Speculative Formulation**: Treebon treats fixed-length token sequences as speculative tasks, while SpecSearch introduces a **flexible bi-level approach**—modeling full reasoning thoughts as high-level tasks and tokens as low-level ones. Unlike Treebon's fixed-length design, SpecSearch leverages LLMs' reasoning capabilities to generates semantically coherent thoughts of dynamic length. (3) **Quality-Preserving Rejection Mechanism**: Treebon rejects a fixed proportion of token sequences using a preset threshold. SpecSearch, instead, scores reasoning thoughts and **adaptively rejects those with lower contextual quality based on the large model's reasoning quality**, enabling finer control and better quality preservation. (4) **Theoretical Guarantee**: Unlike Treebon, which lacks theoretical guarantees, SpecSearch provides **formal assurance that reasoning quality remains uncompromised**, matching that of the large model's outputs.

## 3. Background

**Speculative Sampling in LLM Decoding** We introduce Speculative Sampling (SpS) (Leviathan et al., 2023; Chen et al., 2023a), a decoding technique that accelerates LLM inference while *preserving the target model's distribution*. Given a prefix $c$, a draft model $M_q$, a target model $M_p$, and step size $\gamma$, SpS operates in two phases. (1) **Drafting** $M_q$ autoregressively generates $\gamma$ tokens $x_1, x_2, \ldots, x_\gamma$. (2) **Verification** $M_p$ verifies these tokens in parallel, accepting $x_i$ with probability $\min\left(1, \frac{M_p(x_i|x_{i-1}, \ldots, x_1, c)}{M_q(x_i|x_{i-1}, \ldots, x_1, c)}\right)$. If $x_i$ is rejected, a resampling method generates $\tilde{x}_i$. This process theoretically guarantees alignment with the target model's distribution while significantly enhancing inference speed.

**Tree-Search-Based Reasoning Methods** Tree-search-based reasoning methods formulate tree nodes as intermediate reasoning steps (thoughts) and tree paths as potential solutions to multi-step reasoning problems. They comprise a Thought Generator ($G$), a Thought Evaluator ($V$), and a search algorithm (see Figure 6b in Appendix B). From the root node ($c$, input question), $G$ expands the tree by generating $N$ child nodes (thoughts). $V$ evaluates their quality, guiding the search algorithm. This iterative process constructs a reasoning tree, culminating in a final reasoning path $P$, formed by $z_n, \ldots, z_1, c$. Common search algorithms include Beam Search and MCTS. See Appendix B for details.

## 4. Speculative Search for LLM Reasoning

We begin with an overview of the SpecSearch framework in Section 4.1. Next, we detail the formal procedure underlying SpecSearch, describing the bi-level speculative thought generator in Section 4.2 and the quality-preserving rejection mechanism in Section 4.3. Lastly, we present the theoretical guarantee of SpecSearch in Section 4.4.

### 4.1. Overview of Speculative Search Framework

In this part, we first present several key motivations for our proposed SpecSearch. Then, we describe the overview of SpecSearch as shown in Figure 3.

**Motivation 1: Thought generation dominates computation time in tree-search-based reasoning**, consuming over 91% of total runtime in reasoning (see Figure 1b).

**Motivation 2: Small models can generate high-quality reasoning thoughts.** In multi-step reasoning, some steps are inherently simpler than others. For example, solving $99^2 + 99 + 1$ requires computing $99^2$ ("harder") and $9900 + 1$ ("easier"). Moreover, an analysis of the quantized Qwen2.5-7B-Instruct model shows that over 40% of its reasoning, thoughts outperformed the average reward score of those from the larger Qwen2.5-72B-Instruct model (see Figure 2a). The findings suggest that assigning simple steps to a small model and complex ones to a large model can speed

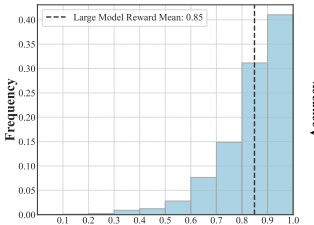
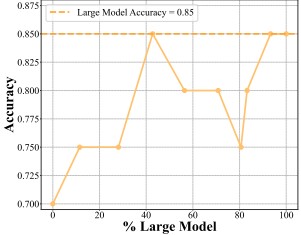

(a) Scores of a Small Model   (b) Large Model Engagement

*Figure 2.* (a) Small models can generate thoughts with high reward scores. (b) Simple large model engagement strategies at the thought level struggle to preserve comparable reasoning quality.

up reasoning without sacrificing accuracy.

**Motivation 3: Simple large model engagement strategies at the thought level struggle to maintain reasoning quality.** Motivated by Motivation 2, we design a simple large-model engagement strategy where a thought evaluator scores the small model's outputs, and the bottom X% (X is a hyperparameter) is reprocessed by the large model for refinement. However, as shown in Figure 2b, maintaining reasoning quality remains challenging when collaboration occurs at the thought level.

**Overview of SpecSearch** Building on the aforementioned key motivations, SpecSearch proposes a bi-level speculative thought generation framework, leveraging both a small model and a large model to efficiently produce high-quality reasoning thoughts. Guided by a thought evaluator, this method operates at both the thought and token levels, integrating seamlessly into any search algorithm as an efficient node expansion module.

The bi-level speculative thought generation follows a **draft-evaluate-reject-correct** paradigm, where the first three stages operate at a coarse-grained thought level, while the final stage refines outputs at a fine-grained token level. Initially, a small model **drafts** multiple reasoning thoughts, which are then **evaluated** by a thought evaluator to **reject** low-quality candidates. Finally, a lossless speculative decoding method **corrects** the rejected thoughts, ensuring robust and accurate reasoning.

### 4.2. Bi-Level Speculative Thought Generator

This section first discusses the advantages of using a small model in collaboration with a large model at both the coarse-grained thought and fine-grained token levels. We then describe the bi-level speculative thought generator. An illustration of the generator is provided in Figure 3. The procedure is summarized in Algorithm 1.

**Advantages** Compared to standard token-level speculative decoding (Xia et al., 2024; Zhang et al., 2024a; Leviathan et al., 2023; Chen et al., 2023a; Li et al., 2024), thought-level speculative execution offers several key advantages. **First**, it leverages the inherent structure of the tree-search-based reasoning framework, where each thought serves as a fundamental unit (i.e., a node) within the reasoning tree. This structure allows for effective utilization of components such as the thought evaluator, enabling seamless integration into the reasoning process. **Second**, since a single thought typically comprises more than fifty tokens (see Table 8 in Appendix I.2), thought-level speculation facilitates coarse-grained collaboration, increasing the number of tokens generated by the small model throughout the search process. This, in turn, can significantly enhance the efficiency of thought generation. Third, it harnesses the reasoning capabilities of small models to generate high-quality thoughts (see Figure 2a). As a result, it carries the potential to maintain or even improve reasoning quality compared to the original large model (see Tabel 1 in Section 5).

**Algorithm Design** We propose the following bi-level speculative thought generator, which follows a draft-evaluate-reject-correct paradigm. First, it drafts multiple reasoning thoughts using a small model, then evaluates the quality of these thoughts and rejects those of low quality. Finally, the rejected thoughts are corrected using a lossless token-level speculative decoding method.

**(1) Drafting Phase at the Thought Level** To leverage the reasoning capability of small models, as shown in Figure 2a, we propose using a small model to efficiently generate multiple reasoning thoughts. These drafted thoughts serve as potential candidates for further evaluation and correction.

**(2) Evaluating Phase at the Thought Level** To evaluate the quality of the generated thoughts, previous speculative decoding methods (Xia et al., 2024; Zhang et al., 2024a) typically use a large model to verify the token sequences within each thought. Verifying thoughts with a large model poses several challenges. **First**, it struggles to capture the intrinsic structure and semantics of reasoning thoughts, leading to potential evaluation inaccuracies. **Second**, while token-level distributions are well understood, preserving thought distributions is far more complex. The exponential growth of possible thoughts makes accurate modeling difficult. **Third**, in tree-search-based reasoning, multiple valid paths can lead to the same answer, creating ambiguity in defining lossless thought generation. A speculative model may generate different reasoning paths than a large model while still being correct. Overall, these challenges significantly limit the accuracy and efficiency of large-model-based verification.

To address these challenges, we propose utilizing the inherent thought evaluator within the existing LLM reasoning framework for accurate thought evaluation. For example, a process reward model (PRM) can be employed to assign a reward score to each thought, offering an accurate evaluation of its quality. This approach addresses the aforementioned challenges and offers several advantages. A detailed discussion is provided in Appendix H.1.

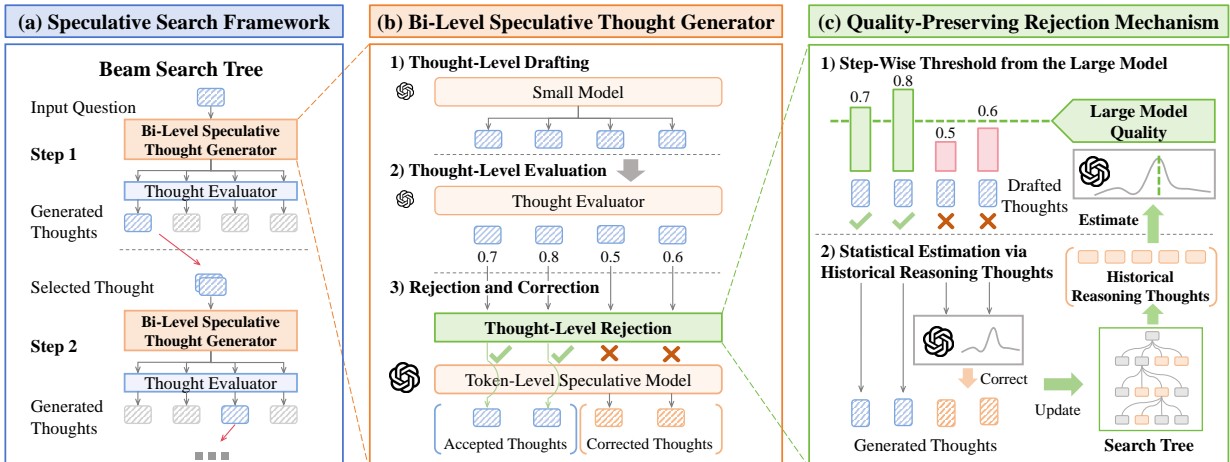

*Figure 3.* Illustration of our proposed SpecSearch. SpecSearch proposes a bi-level speculative thought generator with a quality-preserving rejection mechanism, which significantly accelerates LLM reasoning while preserving comparable quality.

**(3) Rejection Phase at the Thought Level** The primary objective of this phase is to effectively reject generated thoughts that are of lower quality than the large model's outputs—a task made particularly challenging by the lack of access to the large model's outputs. To address this challenge, we propose a novel quality-preserving rejection mechanism, as detailed in Section 4.3.

**(4) Correction Phase at the Token Level** To correct rejected low-quality thoughts, we propose utilizing a lossless token-level speculative decoding method to refine them at a fine-grained token level. By applying lossless speculative decoding, we ensure that the corrected thoughts maintain the same distribution as the large model's outputs. For token-level correction, we propose regenerating the entire thought using a token-level speculative model to replace the rejected one for simplicity. Unless otherwise specified, we use the terms "large model" and "token-level speculative model" interchangeably in the following.

### 4.3. Quality-Preserving Rejection Mechanism

Unlike standard token-level speculative decoding, our approach has the potential to significantly reduce inference latency through thought-level speculation, as discussed in Section 4.2. However, since a reasoning thought consists of more than fifty token-generation steps, a small model is more prone to generating misleading thoughts, as errors can accumulate across multiple token-generation steps. Therefore, a robust thought rejection mechanism is essential to ensure reasoning quality. To address this quality-preserving challenge, we first present several mathematical definitions.

Let $\mathbb{Z}$ be the set of all possible reasoning thoughts, and let $V : \mathbb{Z} \to [0, 1]$ be a process reward model (PRM) that assigns a quality score to each thought. Given a sequence of generated thoughts $z_{k-1}, z_{k-2}, \ldots, z_1$ and an initial condition $c$ (e.g., input question and prompt), a thought generator $G$ samples thoughts from the distribution $G(\cdot|z_{<k})$ over $\mathbb{Z}$.

**Definition 4.1.** (**Quality of Thoughts and Thought Generators**) The quality of a thought $z$ is given by $V(z)$. The reasoning quality of a thought generator $G$ is defined as $\mathbb{E}_{z \sim G(\cdot|z_{<k})} [V(z)]$.

Based on the aforementioned definition, we can compare the quality of our speculative thought generator with that of the large-model-based thought generator. Thus, we present a condition under which our speculative thought generator achieves undegraded quality compared to the large model.

**Definition 4.2.** (**Undegraded Quality Condition**) Let $G_p$ be the large-model-based thought generator, with quality $\mu_p = \mathbb{E}_{z \sim G_p(\cdot|z_{<k})}[V(z)]$. Let $G_s$ be our speculative thought generator. The undegraded quality condition is defined as $\mathbb{E}_{z \sim G_s(\cdot|z_{<k})} [V(z)] \geq \mu_p$.

Based on this condition, we first analyze the quality of our speculative thought generator. In general, the generator operates by first generating $N$ thoughts using a small model, rejecting $M$ low-quality thoughts, and then correcting the rejected $M$ thoughts using a large model (speculative model).

We present an intuitive analysis as follows. In extreme cases, an overly lenient rejection criterion results in $M = 0$, meaning no thoughts are rejected, and the generator relies entirely on the small model. Under these conditions, the acceleration ratio is maximized. However, the quality of the generated thoughts tends to be suboptimal, as the small model is generally less capable than the large model. Conversely, if the rejection criterion is too strict, $M = N$, meaning all thoughts are rejected, reducing the generator to a fully large-model-based approach. While this guarantees reasoning quality, it significantly diminishes acceleration benefits. Therefore, achieving an optimal balance between rejection stringency and computational efficiency is essential to maintain both reasoning quality and acceleration gains.

**Rejection Mechanism Based on Step-Wise Threshold from the Large Model** To implement the aforementioned

**Algorithm 1** Bi-Level Speculative Thought Generator $G_s$

1: **Input:** A sequence of thoughts $z_{<k}$, token-level speculative model $G_p$, small model $G_q$, evaluation model $V$, step-wise threshold $\hat{\beta}^{(k)}$, expansion width $N$, EMA weight $\theta$, a nonparametric estimation method $\Theta$.
2: $\mathcal{T} \leftarrow \left\{ \left(z_k^i, z_{<k}\right) \mid z_k^i \leftarrow G_q(\cdot \mid z_{<k}),\ i = 1, \ldots, N \right\}$
$\qquad\qquad\qquad\qquad\qquad$ ▷ Drafting in Parallel
3: $\mathcal{V} \leftarrow V(\mathcal{T})$ $\qquad\qquad\qquad$ ▷ Evaluating in Parallel
4: Initialize $\mathcal{T}_q \leftarrow \emptyset,\ \mathcal{T}_p \leftarrow \emptyset$
5: **for** $i = 1$ **to** $N$ **do**
6: $\quad$ **if** $\mathcal{V}[i] \geq \hat{\beta}^{(k)}$ **then** $\qquad$ ▷ Rejection Phase
7: $\qquad$ Accept thought: $\mathcal{T}_q \leftarrow \mathcal{T}_q \cup \left\{\mathcal{T}[i]\right\}$
8: $\quad$ **else**
9: $\qquad$ $z_k' \leftarrow G_p\left(\cdot \mid z_{<k}\right)$ $\qquad$ ▷ Correction Phase
10: $\qquad$ $\mathcal{T}_p \leftarrow \mathcal{T}_p \cup \left\{\left(z_k', z_{<k}\right)\right\}$
11: $\quad$ **end if**
12: **end for**
13: $\mathcal{V}_p \leftarrow V(\mathcal{T}_p)$ $\qquad\qquad$ ▷ Evaluating in Parallel
14: $\hat{\beta}^{(k+1)} \leftarrow \theta\hat{\beta}^{(k)} + (1-\theta)\Theta(\mathcal{V}_p)$ ▷ Updating threshold
15: **return** $\hat{\beta}^{(k+1)}, \mathcal{T}_q \cup \mathcal{T}_p$

rejection mechanism, we propose a step-wise threshold-based rejection method. This approach involves establishing a dynamic threshold at each reasoning step, filtering out all thoughts that fall below this threshold to achieve quality-preserving rejection. Intuitively, if the dynamic threshold can be calibrated to reflect the quality of the large model, it is possible to maintain undegraded quality while simultaneously achieving acceleration. This idea is intuitive, and we further validate it theoretically in Section 4.4.

However, at the $k$-th reasoning step, the quality of the large model remains unknown. Fortunately, sampled thought data from the large model is available from previous reasoning steps. Thus, we formulate the step-wise threshold design problem as estimating the large model's quality at the current reasoning step based on historical reasoning thoughts collected during the tree search process.

**Problem Formulation of Step-Wise Threshold Estimation** At the $k$-th reasoning step, we have access to historical data from all previous reasoning steps, along with $M$ corrected thoughts sampled from the large model (speculative model) $G_p$, whose qualities are represented as $\mathcal{V}_p^{(k)} = \left\{V_1^{(k)}, V_2^{(k)}, \ldots, V_M^{(k)}\right\}$. The objective is to utilize this sequence of quality values to predict the large model's thought quality at the next reasoning step and set this estimate as the threshold $\hat{\beta}^{(k+1)}$ for the $(k+1)$-th step.

**Statistical Estimation via Historical Moving Average** To solve the estimation problem, we must leverage the correlation between the large model's quality across different reasoning steps. Without this correlation, the estimation

task would be infeasible. Fortunately, our observations indicate that the quality of the large model's outputs tends to decrease as the reasoning process progresses (see Figure 7 in Appendix I.2). This suggests that at the $(k+1)$-th reasoning step, the quality of the large model's outputs from the previous $k$ steps can serve as an approximate upper bound for the current step's quality estimate. Building on this observation, we propose a two-stage estimation method. First, we estimate the large model's quality at each of the previous $k$ reasoning steps through any non-parametric estimation method. Then, we apply an ensemble weighting technique to these $k$ steps to derive an approximate upper bound for the large model's quality at the $(k+1)$-th reasoning step. Specifically, we incorporate an Exponential Moving Average (EMA) (Klinker, 2011) over the preceding $k$ steps. This approach ensures stable estimation, efficient utilization of historical data, adaptability to dynamic quality shifts, and minimal computational overhead. The update method is as follows: $\hat{\beta}^{(k+1)} = \theta\hat{\beta}^{(k)} + (1-\theta)\Theta\left(\mathcal{V}_p^{(k)}\right)$, where $\theta$ is a hyperparameter controlling the relative importance of historical data $\hat{\beta}^{(k)}$ and current observations $V_i^{(k)}, i = 1, \ldots, M$ in the weighted average. Here, $\Theta$ represents a nonparametric estimation method, such as the sample mean, the confidence upper bound of $\mu_p^{(k)}$, or the maximum value.

In practice, the number of samples from $G_p$ may be limited, leading to highly inaccurate estimates of the large model's quality and, consequently, a significant decline in the quality of generated thoughts. To better utilize historical sample information from the tree search process and improve estimation accuracy, we incorporate data from the small model that passed the rejection phase. We treat these samples as an approximate upper-bound estimate of the large model's quality and integrate them into our estimation framework.

### 4.4. Theoretical Guarantee of Undegraded Quality

We provide the following theoretical analysis to demonstrate that our SpecSearch can guarantee undegraded reasoning quality. We provide detailed proof in Appendix A.

In this section, we analyze the integration of SpecSearch with the beam search algorithm, which operates with a maximum reasoning depth of $K$. At each step, the algorithm generates $N$ candidate thoughts and selects the best one.

Let $G_p$ and $G_q$ denote the large model (speculative model) and the small model. Due to the complexity of large language models, rigorous mathematical reasoning is challenging. To simplify the mathematical derivation, we assume that, at the $k$-th reasoning step, the qualities of thoughts generated by $G_p$ and $G_q$ independently follow normal distributions, $\mathcal{N}\left(\mu_p^{(k)}, \sigma_p^{(k)}\right)$ and $\mathcal{N}\left(\mu_q^{(k)}, \sigma_q^{(k)}\right)$, respectively, where $\mu_p^{(k)} \geq \mu_q^{(k)} > 0$. This assumption is commonly used in data science (Gopinath, 1998; Zhang, 2010).

Denote our speculative thought generator with threshold $\beta^{(k)}, k = 1, 2, \ldots, K$ as $G_s \left( \{ \beta^{(k)} \}_{k=1}^K \right)$, where $K$ is the maximum number of reasoning steps. Note that we use $\beta^{(k)}$ to denote a general threshold, while $\hat{\beta}^{(k)}$ represents the estimate of the threshold obtained using our estimation method. The following theorem guarantees that, under ideal conditions, as long as the threshold meets or exceeds the quality of the large model, our generator ensures the undegraded quality condition defined in Definition 4.2.

**Theorem 4.3.** (***Quality-Preserving Condition on the Threshold***) *The generator* $G_s \left( \{ \beta^{(k)} \}_{k=1}^K \right)$ *preserves undegraded quality if the following condition holds:* $\beta^{(k)} \geq \mu_p^{(k)}, \ \forall k = 1, 2, \ldots, K.$

This theorem provides a quality-preserving condition on the threshold in our designed speculative thought generator. Specifically, if the threshold estimation method proposed in Section 4.3 satisfies this condition, then our SpecSearch guarantees undegraded quality.

We then make the following quality-descending assumption based on our observations in Figure 7 in Appendix I.2.

**Assumption 4.4.** (**Descending Quality and Bounded Variance**) At the $k$-th step in the beam search algorithm, which selects the candidate with optimal quality, we assume that $\mu_p^{(k)} \leq \gamma \mu_p^{(k-1)}, \ \forall k = 1, 2, \ldots, K$, where $\gamma < 1$ is the decay factor. We further assume that $\sigma_p^{(k)} \leq \sigma_c, \ \forall k = 1, 2, \ldots, K$, where $\sigma_c > 0$ is a constant.

Building upon this assumption, the following theorem establishes a lower bound on the probability that our speculative thought generator preserves quality at each reasoning step.

**Theorem 4.5.** (***Probability Bound for Step-Wise Quality-Preserving***) *Consider a speculative thought generator* $G_s \left( \{ \beta^{(k)} \}_{k=1}^K \right)$. *Given that weight* $\theta \geq \gamma$ *and at step* $k \geq 1$, *the generator* $G_s$ *preserves undegraded quality, the lower bound for the probability that at step* $k + 1$ *it also preserves undegraded quality is given by:*

$$P \left( \hat{\beta}^{(k+1)} \geq \mu_p^{(k+1)} \mid \hat{\beta}^{(k)} \geq \mu_p^{(k)} \right) \geq$$
$$\frac{\left[ \frac{1-\gamma}{\gamma} \mu_p^{(k+1)} \right]^2}{\left[ \frac{1-\gamma}{\gamma} \mu_p^{(k+1)} \right]^2 + \left( \frac{1}{N+1} + \frac{2}{N+2} \right)(\sigma_c)^2}. \quad (1)$$

Furthermore, for a beam search algorithm with up to $K$ reasoning steps, we derive a lower bound on the probability that our speculative thought generator maintains undegraded quality, as stated in the following theorem.

**Theorem 4.6.** (***Probability Bound for Quality-Preserving***) *For a speculative thought generator* $G_s \left( \{ \beta^{(k)} \}_{k=1}^K \right)$ *with a maximum of $K$ reasoning steps, where* $K \in \mathbb{N}$, *and weight* $\theta \geq \gamma$, *the lower bound on the probability of $G_s$ preserving*

*undegraded quality is given by:*

$$P \left( \hat{\beta}^{(k)} \geq \mu_p^{(k)}, 1 \leq k \leq K \right) \geq$$
$$\left( 1 - \frac{1}{2^{N+1}} \right) \prod_{k=1}^{K-1} \left[ \frac{\left[ \frac{1-\gamma}{\gamma} \mu_p^{(k+1)} \right]^2}{\left[ \frac{1-\gamma}{\gamma} \mu_p^{(k+1)} \right]^2 + \left( \frac{1}{N+1} + \frac{2}{N+2} \right)(\sigma_c)^2} \right]. \quad (2)$$

This probability bound increases monotonically with respect to the sample size $N$. Furthermore, as $N \to \infty$, the lower bound approaches 1, implying that our speculative generator can achieve higher reasoning quality by generating more samples during the drafting phase. A detailed discussion of this result is provided in Appendix A.5.

## 5. Experiments

Our experiments have four main parts. **Experiment 1.** We evaluate the performance of SpecSearch and the baselines on different datasets and LLMs. **Experiment 2.** We evaluate the generalization performance of SpecSearch across different search algorithms and thought evaluators. **Experiment 3.** We conduct carefully designed ablation studies to demonstrate the effectiveness of SpecSearch. **Experiment 4.** We perform a visualization analysis of SpecSearch to provide further insight into SpecSearch.

**Experimental Setup** We use quantized Qwen2.5-72B-Instruct and Qwen2.5-7B-Instruct (Team, 2024) as large and small models, respectively, along with quantized Llama3-70B-Instruct and Llama3-8B-Instruct (Dubey et al., 2024). Unless stated otherwise, experiments follow OpenR (Wang et al., 2024a) settings: tree width of 6, tree depth of 50, MATH-psa as the process reward model (PRM), Qwen models as the main LLMs, and beam search as the main search algorithm. Throughout all experiments, we set the EMA weight $\theta$ in SpecSearch to 0.9

**Baselines** This study aims to accelerate thought generation in reasoning trees without modifying search algorithms or prompting techniques. Thus, we compare our method with two baselines: (1) AR, the original ToT method using autoregressive decoding with a large model, and (2) SpS, a state-of-the-art (SOTA) lossless speculative decoding approach. Details are in Appendix D.

**Datasets** We use two well-established mathematical problem datasets, GSM8K (Cobbe et al., 2021) and MATH (Hendrycks et al., 2021), to evaluate the acceleration performance of the proposed framework. GSM8K contains high-quality elementary mathematics word problems, while MATH comprises advanced high school competition-level math problems. Due to the long testing times of tree-search-based reasoning methods, we randomly select 100 samples from both the GSM8K and MATH datasets for evaluation.

**Evaluation Metrics.** We use two widely-used metrics, *accuracy* and *speedup*, to compare our method's performance with that of the baselines. We define the *accuracy* by the

*Table 1.* The results demonstrate that SpecSearch significantly accelerates LLM reasoning with comparable reasoning accuracy.

| Models | Methods | Dataset MATH-100 | | | | GSM8K-100 | | | |
|---|---|---|---|---|---|---|---|---|---|
| | | Reasoning Accuracy (%) ↑ | Average Inference Latency (s) ↓ | Speedup (vs AR) ↑ | Speedup (vs SpS) ↑ | Reasoning Accuracy (%) ↑ | Average Inference Latency (s) ↓ | Speedup (vs AR) ↑ | Speedup (vs SpS) ↑ |
| Qwen | AR | 87.00 | 275.78 | NA | 0.51 | 97.00 | 138.24 | NA | 0.50 |
| | SpS | 88.00 | 141.55 | 1.95 | NA | 97.00 | 69.43 | 1.99 | NA |
| | SpecSearch (Ours) | 87.00 | **82.35** | **3.35** | **1.72** | 96.00 | **48.18** | **2.87** | **1.44** |
| | Methods | Reasoning Accuracy (%) ↑ | Average Inference Latency (s) ↓ | Speedup (vs AR) ↑ | Speedup (vs SpS) ↑ | Reasoning Accuracy (%) ↑ | Average Inference Latency (s) ↓ | Speedup (vs AR) ↑ | Speedup (vs SpS) ↑ |
| Llama | AR | 62.00 | 170.84 | NA | 0.79 | 87.00 | 90.04 | NA | 0.71 |
| | SpS | 61.00 | 134.34 | 1.27 | NA | 86.00 | 64.29 | 1.40 | NA |
| | SpecSearch (Ours) | 64.00 | **129.65** | **1.32** | **1.04** | 88.00 | **45.34** | **1.99** | **1.42** |

percentage of correct predictions. We define the *speedup* by the ratio of the baseline's latency to our approach's latency.

**Experiment 1. Main Evaluation** We evaluate SpecSearch against two competitive baselines on two math datasets using the Qwen and Llama models. Table 1 highlights three key findings. Moreover, we provide additional evaluation on four more distinct dataset categories, including the full GSM8K, AIME, Olympiad Bench, and a code-generation benchmark, in Appendix I.1.

(1) **High Speedup** SpecSearch consistently outperforms all baselines, achieving up to 1.72× speedup over SpS and 3.35× over AR on MATH-100 with Qwen. (2) **Broad Compatibility** Our method accelerates both Llama and Qwen models, demonstrating strong adaptability across LLMs. (3) **Superior Reasoning Ability** On Llama, SpecSearch surpasses baselines in reasoning accuracy on MATH-100 and GSM8K-100, highlighting the strong ability of our SpecSearch to effectively collaborate the small and large models to maintain reasoning quality.(4) **Accuracy Degradation Analysis** We conduct a case study to explore the reasons behind the accuracy degradation of SpecSearch on GSM8K-100. The results in Appendix I.3 show that the degradation primarily arises from certain misleading thoughts that deceive the PRM.

**Experiment 2. Generalization** We evaluate SpecSearch's generalization across different search algorithms and thought evaluators on GSM8K-100. Due to limited space, we defer results on MATH-100 to Appendix I.4.

**Search Algorithms** To demonstrate the broad applicability of SpecSearch, we apply it to two distinct search algorithms: beam search and MCTS. We compare SpecSearch against AR and SpS on both algorithms. As shown in Table 2, SpecSearch significantly outperforms the baselines, reducing inference latency while maintaining comparable accuracy. These results highlight both its efficiency and its adaptability across different search algorithms.

**Different Thought Evaluators** To evaluate the generalization of SpecSearch across thought evaluators, we test

it with two PRMs—Math-Shepherd (Wang et al., 2024c) and MATH-psa (Wang et al., 2024a)—on beam search. As shown in Table 2, SpecSearch maintains nearly the same accuracy while significantly accelerating inference across different PRMs, achieving up to 2.12× speedup. This demonstrates its strong adaptability to various evaluators.

**Experiment 3. Ablation Study** As the MATH dataset is harder than the GSM8K dataset, we perform an ablation study and sensitivity analysis on the MATH dataset. Specifically, we further randomly sample 50 problems from MATH-100, called MATH-50.

**Contribution of Each Component** To assess the effectiveness of each component, we conduct an ablation study. For the **Evaluation Module** in SpecSearch, we replace PRM evaluation with evaluation via the log probabilities of a large model. For the **Rejection Module**, we compare three variations: SpecSearch with Fixed Large Model Engagement (SpecSearch w/ FLME), SpecSearch with Fixed Threshold (SpecSearch w/ FT), and SpecSearch with Random Rejection (SpecSearch w/ RR). SpecSearch w/ FLME implements a simple collaboration strategy between large and small models. SpecSearch w/ FT replaces the step-wise threshold estimation with a fixed threshold in the rejection process. SpecSearch w/ RR randomly rejects 50% of the small model's generated thoughts. As shown in Table 3, our evaluation and rejection modules are both essential for preserving reasoning quality, suggesting that each component in our proposed SpecSearch are important for its significant performance improvement.

**Sensitivity Analysis** (1) **The EMA Weight** $\theta$**.** We analyze the sensitivity of SpecSearch to the EMA weight $\theta$. Due to limited space, we defer results to Appendix I.6. The results in Table 11 in Appendix I.6 show that SpecSearch achieves similar average performance across a wide range of $\theta$. (2) **Draft Model's Size** We have investigated SpecSearch's performance using multiple small draft models. The results in Table 10 in Appendix I.5 reveal that SpecSearch achieves speedups ranging from 2.18× to 2.87×, underscoring its acceleration capabilities across diverse small-model settings.

*Table 2.* The results demonstrate the Broad Compatibility of Our SpecSearch with different search algorithms and PRMs.

| Search Algorithms | Beam Search | | | | MCTS | | | |
|---|---|---|---|---|---|---|---|---|
| Methods | Reasoning Accuracy (%) ↑ | Average Inference Latency (s) ↓ | Speedup (vs AR) ↑ | Speedup (vs SpS) ↑ | Reasoning Accuracy (%) ↑ | Average Inference Latency (s) ↓ | Speedup (vs AR) ↑ | Speedup (vs SpS) ↑ |
| AR | 97.00 | 138.24 | NA | 0.50 | 98.00 | 256.17 | NA | 0.51 |
| SpS | 97.00 | 69.43 | 1.99 | NA | 98.00 | 129.74 | 1.97 | NA |
| SpecSearch (Ours) | 96.00 | **48.18** | **2.87** | **1.44** | 98.00 | **98.16** | **2.61** | **1.32** |
| **PRMs** | Math-Shepherd | | | | Math-psa | | | |
| Methods | Reasoning Accuracy (%) ↑ | Average Inference Latency (s) ↓ | Speedup (vs AR) ↑ | Speedup (vs SpS) ↑ | Reasoning Accuracy (%) ↑ | Average Inference Latency (s) ↓ | Speedup (vs AR) ↑ | Speedup (vs SpS) ↑ |
| AR | 96.00 | 124.76 | NA | 0.51 | 97.00 | 138.24 | NA | 0.50 |
| SpS | 95.00 | 64.17 | 1.94 | NA | 97.00 | 69.43 | 1.99 | NA |
| SpecSearch (Ours) | 94.00 | **30.32** | **4.11** | **2.12** | 96.00 | **48.18** | **2.87** | **1.44** |

*Table 3.* The results demonstrate that each component within Spec-Search is significant for maintaining reasoning accuracy.

| Dataset | MATH-50 | | |
|---|---|---|---|
| Methods | Reasoning Accuracy (%) ↑ | Average Inference Latency (s) ↓ | Speedup (vs AR) ↑ |
| AR | 88.00 | 256.05 | NA |
| SD | 90.00 | 132.68 | 1.93 |
| SpecSearch (Ours) | 88.00 | 70.63 | 3.63 |
| Evaluation Module | | | |
| SpecSearch w LMV | 78.00 | 172.26 | 1.49 |
| Rejection Module | | | |
| SpecSearch w FT | 80.00 | 68.84 | 3.72 |
| SpecSearch w RR | 80.00 | 99.73 | 2.57 |
| SpecSearch w FLME | 84.00 | 105.25 | 2.43 |

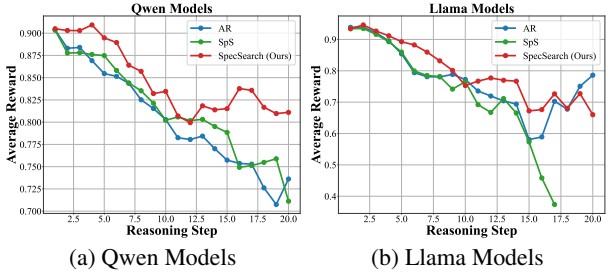

(a) Qwen Models      (b) Llama Models

*Figure 4.* To verify that our method preserves comparable reward scores for reasoning thoughts, we visualize the average reward scores at each reasoning step during the tree search process.

**Experiment 4. Visualization Analysis** To evaluate whether our SpecSearch can preserve comparable reward scores for reasoning thoughts, we visualize the average reward scores at each reasoning step during the tree search process for SpecSearch and the baselines on the MATH-100 dataset. As shown in Figure 4, SpecSearch achieves reward scores comparable to those of the large model across all reasoning steps. This result highlights SpecSearch's ability to significantly accelerate inference while maintaining comparable reasoning quality to the large model.

## 6. Conclusion

We propose Speculative Search (SpecSearch), a framework that accelerates reasoning by enabling a small model to generate speculative thoughts with a large model at both thought and token levels. With a quality-preserving rejection mechanism, SpecSearch theoretically maintains reasoning quality comparable to the large model. Experiments show up to 2.12× speedup while preserving high reasoning quality.

## Acknowledgements

This work was supported in part by National Key R&D Program of China under contract 2022ZD0119801, National Nature Science Foundations of China grants U23A20388 and 62021001. This work was supported in part by Huawei as well. We would like to thank all the anonymous reviewers for their insightful comments.

## Impact Statement

This paper presents work whose goal is to advance the field of Machine Learning. There are many potential societal consequences of our work, none of which we feel must be specifically highlighted here.

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

# A. Theoretical Analysis

In this section, we provide proof of the theorems in the main paper along with further discussions.

To facilitate analytical clarity, our analysis is confined to the case that the sample mean serves as the nonparametric estimation method, i.e.

$$\hat{\beta}^{(k+1)} = \theta \hat{\beta}^{(k)} + \frac{1-\theta}{M+1} \sum_{i=1}^{M+1} V_i^{(k)}, \tag{3}$$

where we assume that the large model generates one more thought to avoid the case where $M = 0$. We present further discussion about those settings in Appendix A.5.3 and Appendix A.5.2.

## A.1. Proof for Theorem 4.3

**Lemma A.1.** *Let $\varphi(x)$ and $\Phi(x)$ denote the probability density function (PDF) and cumulative distribution function (CDF) of the standard normal, respectively. Then for any $x \in \mathbb{R}$, we have*

$$\varphi(x) - x(1 - \Phi(x)) > 0. \tag{4}$$

*distribution*

*Proof.* Notice that

$$
\begin{aligned}
1 - \Phi(x) &= \int_x^\infty \varphi(t) dt \\
&= \int_x^\infty \frac{1}{t} \cdot t\varphi(t) dt \\
&= -\int_x^\infty \frac{1}{t} \cdot d\varphi(t) \\
&= -\frac{1}{t} \varphi(t) \Big|_{t=x}^\infty + \int_x^\infty \varphi(t) d\left(\frac{1}{t}\right) \\
&= \frac{1}{x} \varphi(x) - \int_x^\infty \varphi(t) \frac{1}{t^2} dt \\
&< \frac{1}{x} \varphi(x).
\end{aligned}
\tag{5}
$$

Then we have

$$\varphi(x) - x(1 - \Phi(x)) > 0. \tag{6}$$

$\square$

Then we prove Theorem 4.3 as follows.

*Proof.* At step $k$, let the qualities of $N$ thoughts obtained from the small model $G_q$ at the $k$-th step be denoted as $\hat{V}_i^{(k)}$, $i = 1, 2, \ldots, N$, and the threshold of the generator as $\beta^{(k)}$. Among those, $U$ thoughts with qualities $\hat{V}_{i_l}^{(k)}, l = 1, 2, \ldots, U$ are retained with $\hat{V}_{i_l}^{(k)} \geq \beta$. The rest of $M = N - U$ thoughts are refined by large model $G_p$ with qualities $V_i^{(k)}, i = 1, 2, \ldots, N - U$, along with an additional thought generated with quality $V_{N-U+1}^{(k)}$ by the target model. Then the probability that a sample passes the threshold $\beta^{(k)}$ is given by

$$p_s = 1 - \Phi\left(\frac{\beta^{(k)} - \mu_q^{(k)}}{\sigma_q^{(k)}}\right). \tag{7}$$

Thus, the number of passing thoughts $U$ follows a binomial distribution over $N$ trials:

$$P(U = u) = \binom{N}{u} p_s^u (1 - p_s)^{N-u}. \tag{8}$$

For the qualities of passing thoughts $\hat{V}_{i_k}$, their distribution is a truncated normal distribution(Burkardt, 2014). The expected quality is computed as

$$\mu_q' = \mathbb{E}\left[\hat{V}_{i_l}^{(k)}\right] = \mu_q^{(k)} + \sigma_q^{(k)} \frac{\varphi\left(\dfrac{\beta^{(k)} - \mu_q^{(k)}}{\sigma_q^{(k)}}\right)}{1 - \Phi\left(\dfrac{\beta^{(k)} - \mu_q^{(k)}}{\sigma_q^{(k)}}\right)}. \tag{9}$$

Let the mean quality of the new batch of solutions be denoted as

$$\overline{V}^{(k)} = \frac{1}{N+1}\left(\sum_{l=1}^{U} \hat{V}_{i_l}^{(k)} + \sum_{j=1}^{N-U+1} V_j^{(k)}\right). \tag{10}$$

By the law of total expectation(Dekking et al., 2006), we have

$$\begin{aligned}
\mathbb{E}\left[\overline{V}^{(k)}\right] &= \mathbb{E}\left[\mathbb{E}\left[\overline{V}^{(k)}|U\right]\right] \\
&= \sum_{u=0}^{n} P(U = u)\mathbb{E}\left[\overline{V}^{(k)}|U = u\right] \\
&= \sum_{u=0}^{n} P(U = u)\mathbb{E}\left[\frac{1}{N+1}\left(\sum_{l=1}^{U} \hat{V}_{i_l}^{(k)} + \sum_{j=1}^{N-u+1} V_j^{(k)}\right) \Big| U = u\right] \\
&= \sum_{u=0}^{n} P(U = u)\left[\frac{1}{N+1}\left(\sum_{l=1}^{U} \mathbb{E}[\hat{V}_{i_l}^{(k)}] + \sum_{j=1}^{N-u+1} \mathbb{E}[V_j^{(k)}]\right) \Big| U = u\right] \\
&= \sum_{u=0}^{n} P(U = u)\left[\frac{u}{N+1}\mu_q' + \frac{N-u+1}{N+1}\mu_p^{(k)}\Big| U = u\right] \\
&= \sum_{u=0}^{n} P(U = u)\left[\mu_p^{(k)} + \frac{u}{N+1}(\mu_q' - \mu_p^{(k)})\Big| U = u\right] \\
&= \mu_p^{(k)} + \frac{\mu_q' - \mu_p^{(k)}}{N+1}\sum_{u=0}^{N} uP(U = u) \\
&= \mu_p^{(k)} + \frac{\mu_q' - \mu_p^{(k)}}{N+1}\mathbb{E}[U] \\
&= \mu_p^{(k)} + \frac{Np_s}{N+1}(\mu_q' - \mu_p^{(k)}). 
\end{aligned} \tag{11}$$

Let

$$h(x) = \frac{\varphi(x)}{1 - \Phi(x)}. \tag{12}$$

and we have

$$h'(x) = \frac{-x\varphi(x)(1 - \Phi(x)) + \varphi^2(x)}{(1 - \Phi(x))^2} = \frac{\varphi(x)(\varphi(x) - x(1 - \Phi(x)))}{(1 - \Phi(x))^2}. \tag{13}$$

According to Lemma A.1, $\varphi(x) - x(1 - \Phi(x)) > 0$, i.e. $h(x) \geq x$, so the function $h(x)$ is monotonically increasing. Therefore for any $k \geq 1$,

$$\mu'_q = \mu_q^{(k)} + \sigma_q^{(k)} h\left(\frac{\beta^{(k)} - \mu_q^{(k)}}{\sigma_q^{(k)}}\right) \geq \mu_q^{(k)} + \sigma_q^{(k)} h\left(\frac{\mu_p^{(k)} - \mu_q^{(k)}}{\sigma_q^{(k)}}\right) \geq \mu_q^{(k)} + \sigma_q^{(k)} \frac{\mu_p^{(k)} - \mu_q^{(k)}}{\sigma_q^{(k)}} = \mu_p^{(k)}. \tag{14}$$

Substituting into the (11), we obtain that

$$\mathbb{E}\left[\overline{V}^{(k)}\right] \geq \mu_p^{(k)}, \quad \forall k \geq 1, \tag{15}$$

which is the definition of a lossless thought generator.

$\square$

## A.2. Further Discussion on Theorem 4.3

Additionally, we provide the necessary and sufficient conditions for losslessness in the following proposition.

**Proposition A.2.** (*Necessary And Sufficient Lossless Threshold Condition*) *The generator $G_s(\beta)$ is lossless **if and only if** for any reasoning step $k \geq 1$,*

$$\beta^{(k)} \geq \mu_q^{(k)} + \alpha^{(k)} \sigma_q^{(k)}, \tag{16}$$

*where $\alpha^{(k)}$ is the solution to the equation*

$$\frac{\varphi(x)}{1 - \Phi(x)} = \frac{\mu_p^{(k)} - \mu_q^{(k)}}{\sigma_q^{(k)}}. \tag{17}$$

*Proof.* As shown in the proof for Theorem 4.3, we have

$$\mathbb{E}\left[\overline{V}^{(k)}\right] = \mu_p^{(k)} + \frac{N p_s}{N + 1}(\mu'_q - \mu_p^{(k)}). \tag{18}$$

The condition is equivalent to

$$\beta^{(k)} \geq \mu_q^{(k)} + \alpha^{(k)} \sigma_q^{(k)} \iff \frac{\beta^{(k)} - \mu_q^{(k)}}{\sigma_q^{(k)}} \geq \alpha^{(k)}, \quad \forall k \geq 1. \tag{19}$$

By the monotonicity of the function $h(x) = \frac{\varphi(x)}{1 - \Phi(x)}$, inequality (19) is equivalent to

$$\frac{\varphi\left(\frac{\beta^{(k)} - \mu_q^{(k)}}{\sigma_q^{(k)}}\right)}{1 - \Phi\left(\frac{\beta^{(k)} - \mu_q^{(k)}}{\sigma_q^{(k)}}\right)} \geq h(\alpha^{(k)}) = \frac{\mu_p^{(k)} - \mu_q^{(k)}}{\sigma_q^{(k)}}, \quad \forall k \geq 1. \tag{20}$$

Rearranging, we obtain

$$\mu'_q - \mu_p^{(k)} = \mu_q^{(k)} + \sigma_q^{(k)} \frac{\varphi\left(\frac{\beta^{(k)} - \mu_q^{(k)}}{\sigma_q^{(k)}}\right)}{1 - \Phi\left(\frac{\beta^{(k)} - \mu_q^{(k)}}{\sigma_q^{(k)}}\right)} - \mu_p^{(k)} \geq 0, \quad \forall k \geq 1. \tag{21}$$

Substituting into the (18), we obtain that (21) is equivalent to

$$\mathbb{E}\left[\overline{V}^{(k)}\right] \geq \mu_p^{(k)}, \quad \forall k \geq 1. \tag{22}$$

This is equivalent to the definition of a lossless thought generator.

$\square$

### A.3. Proof for Theorem 4.5

Let the condition be denoted as $\mathcal{C} = \left\{\hat{\beta}^{(k)} \geq \mu_p^{(k)}\right\}$, and we have

$$
\begin{aligned}
\mathbb{E}\left[\hat{\beta}^{(k+1)} \mid \mathcal{C}\right] &= \theta \mathbb{E}\left[\hat{\beta}^{(k)} \mid \mathcal{C}\right] + (1-\theta)\mathbb{E}\left[\overline{V}^{(k)} \mid \mathcal{C}\right] \\
&\geq \theta \mu_p^{(k)} + (1-\theta)\mu_p^{(k)} \\
&= \mu_p^{(k)} \geq \frac{1}{\gamma}\mu_p^{(k+1)}
\end{aligned}
\tag{23}
$$

Thus,

$$\mathbb{E}\left[\hat{\beta}^{(k+1)} \mid \mathcal{C}\right] - \mu_p^{(k+1)} \geq \frac{1-\gamma}{\gamma}\mu_p^{(k+1)} \tag{24}$$

Then according to condition variance decomposition (Dekking et al., 2006), we have

$$
\begin{aligned}
Var\left[\hat{\beta}^{(k+1)} \mid \mathcal{C}\right] &= Var\left[\mathbb{E}\left[\hat{\beta}^{(k+1)} \mid \mathcal{C}, \hat{\beta}^{(k)}\right] \mid \mathcal{C}\right] + \mathbb{E}\left[Var\left[\hat{\beta}^{(k+1)} \mid \mathcal{C}, \hat{\beta}^{(k)}\right] \mid \mathcal{C}\right] \\
&= I + II
\end{aligned}
\tag{25}
$$

where $I = Var\left[\mathbb{E}\left[\hat{\beta}^{(k+1)} \mid \mathcal{C}, \hat{\beta}^{(k)}\right] \mid \mathcal{C}\right]$ and $II = \mathbb{E}\left[Var\left[\hat{\beta}^{(k+1)} \mid \mathcal{C}, \hat{\beta}^{(k)}\right] \mid \mathcal{C}\right]$. For $I$, since

$$
\begin{aligned}
\mathbb{E}\left[\hat{\beta}^{(k+1)} \mid \mathcal{C}, \hat{\beta}^{(k)}\right] &= \mathbb{E}\left[\hat{\beta}^{(k+1)} \mid \mathcal{C}, \hat{\beta}^{(k)}\right] \\
&= \mathbb{E}\left[\theta\hat{\beta}^{(k)} + (1-\theta)\overline{V}^{(k)} \mid \mathcal{C}, \hat{\beta}^{(k)}\right] \\
&= \theta\mathbb{E}\left[\hat{\beta}^{(k)} \mid \mathcal{C}, \hat{\beta}^{(k)}\right] + (1-\theta)\mathbb{E}\left[\overline{V}^{(k)} \mid \mathcal{C}, \hat{\beta}^{(k)}\right] \\
&= \theta\hat{\beta}^{(k)} + (1-\theta)\mu_p^{(k)}.
\end{aligned}
\tag{26}
$$

and therefore we have,

$$I = Var\left[\theta\hat{\beta}^{(k)} + (1-\theta)\mu_p^{(k)} \mid \mathcal{C}\right] = \theta^2 Var\left[\hat{\beta}^{(k)} \mid \mathcal{C}\right]. \tag{27}$$

For $II$, now that

$$Var\left[\hat{\beta}^{(k+1)} \mid \mathcal{C}, \hat{\beta}^{(k)}\right] = Var\left[\theta\hat{\beta}^{(k)} + (1-\theta)\overline{V}^{(k)} \mid \mathcal{C}, \hat{\beta}^{(k)}\right]$$

$$= (1-\theta)^2 Var\left[\overline{V}^{(k)} \mid \mathcal{C}, \hat{\beta}^{(k)}\right]$$

$$= (1-\theta)^2 \left( Var\left[\mathbb{E}\left[\overline{V}^{(k)} \mid \mathcal{C}, \hat{\beta}^{(k)}, U^{(k)}\right] \mid \mathcal{C}, \hat{\beta}^{(k)}\right] \right. \tag{28}$$

$$\left. + \mathbb{E}\left[Var\left[\overline{V}^{(k)} \mid \mathcal{C}, \hat{\beta}^{(k)}, U^{(k)}\right] \mid \mathcal{C}, \hat{\beta}^{(k)}\right] \right)$$

$$= (1-\theta)^2 \left( II_1 + II_2 \right), \tag{29}$$

where $U^{(k)}$ is the number retained draft thought. We find that

$$II_1 = Var\left[\mathbb{E}\left[\overline{V}^{(k)} \mid \mathcal{C}, \hat{\beta}^{(k)}, U^{(k)}\right] \mid \mathcal{C}, \hat{\beta}^{(k)}\right]$$

$$= Var\left[\mu_p^{(k)} \mid \mathcal{C}, \hat{\beta}^{(k)}\right] = 0, \tag{30}$$

and

$$Var\left[\overline{V}^{(k)} \mid \mathcal{C}, \hat{\beta}^{(k)}, U^{(k)}\right]$$

$$= Var\left[\frac{1}{N - U^{(k)} + 1} \sum_{i=1}^{N-U^{(k)}+1} V_i^{(k)} \mid \mathcal{C}, \hat{\beta}^{(k)}, U^{(k)}\right]$$

$$= \frac{\left(\sigma_p^{(k)}\right)^2}{N - U^{(k)} + 1}. \tag{31}$$

Since function $g(x) = \dfrac{1}{N - x + 1}$ is a concave function, therefore according to Jensen's Inequality (Dekking et al., 2006),

$$II_2 = \left(\sigma_p^{(k)}\right)^2 \mathbb{E}\left[g(U^{(k)}) \mid \mathcal{C}, \hat{\beta}^{(k)}\right]$$

$$\leq \left(\sigma_p^{(k)}\right)^2 g\left(\mathbb{E}\left[U^{(k)} \mid \mathcal{C}, \hat{\beta}^{(k)}\right]\right)$$

$$= \frac{\left(\sigma_p^{(k)}\right)^2}{N - \mathbb{E}\left[U^{(k)} \mid \mathcal{C}, \hat{\beta}^{(k)}\right] + 1}$$

$$= \frac{\left(\sigma_p^{(k)}\right)^2}{N - Np_s + 1} \tag{32}$$

where $p_s$ is defined in (7). Given condition where $\hat{\beta}^{(k)} \geq \mu_p^{(k)}$, then

$$p_s = 1 - \Phi\left(\frac{\beta^{(k)} - \mu_q^{(k)}}{\sigma_q^{(k)}}\right) \leq 1 - \Phi\left(\frac{\mu_p^{(k)} - \mu_q^{(k)}}{\sigma_q^{(k)}}\right) \leq 1 - \Phi\left(\frac{\mu_q^{(k)} - \mu_q^{(k)}}{\sigma_q^{(k)}}\right) = \frac{1}{2}.$$

Therefore,

$$II_2 \leq \frac{\left(\sigma_p^{(k)}\right)^2}{N - \frac{1}{2}N + 1} = \frac{2}{N+2}\left(\sigma_p^{(k)}\right)^2 \tag{33}$$

Overall, we can find the recursive expression of variance of $\hat{\beta}^{(k)}$:

$$Var\left[\hat{\beta}^{(k+1)} \mid \mathcal{C}\right] \leq \theta^2 Var\left[\hat{\beta}^{(k)} \mid \mathcal{C}\right] + \frac{2(1-\theta)^2}{N+2}\left(\sigma_p^{(k)}\right)^2 \tag{34}$$

Now that $\left(\sigma_p^{(k)}\right)^2 \leq (\sigma_c)^2$, we can derive that

$$
\begin{aligned}
Var\left[\hat{\beta}^{(k+1)} \mid \mathcal{C}\right] &\leq \theta^2 Var\left[\hat{\beta}^{(k)} \mid \mathcal{C}\right] + \frac{2(1-\theta)^2}{N+2}(\sigma_c)^2 \\
&\cdots \\
&\leq \theta^{2k} Var\left[\hat{\beta}^{(1)} \mid \mathcal{C}\right] + \frac{2(1-\theta)^2}{N+2}\sum_{i=1}^{k}\theta^{2i-2}(\sigma_c)^2 \\
&\leq \left(\frac{\theta^{2k}}{N+1} + \frac{2}{N+2}\frac{(1-\theta)^2\left(1-\theta^{2k}\right)}{1-\theta^2}\right)(\sigma_c)^2 \\
&= \left(\frac{\theta^{2k}}{N+1} + \frac{2}{N+2}\frac{(1-\theta)\left(1-\theta^{2k}\right)}{1+\theta}\right)(\sigma_c)^2 \\
&\leq \left(\frac{1}{N+1} + \frac{2}{N+2}\frac{(1-\gamma)(1-\gamma^{2k})}{1+\gamma}\right)(\sigma_c)^2 \\
&\leq \left(\frac{1}{N+1} + \frac{2}{N+2}\right)(\sigma_c)^2.
\end{aligned}
\tag{35}
$$

By Cantelli's inequality (Wikipedia contributors, 2024),

$$
\begin{aligned}
P\left(\hat{\beta}^{(k+1)} \leq \mu_p^{(k+1)}\bigg|\mathcal{C}\right) &= P\left(\hat{\beta}^{(k+1)} - \mathbb{E}\left[\hat{\beta}^{(k+1)}|\mathcal{C}\right] \leq -\left(\mathbb{E}\left[\hat{\beta}^{(k+1)}|\mathcal{C}\right] - \mu_p^{(k+1)}\right)\bigg|\mathcal{C}\right) \\
&\leq \left(1 + \frac{\left(\mathbb{E}\left[\hat{\beta}^{(k+1)}|\mathcal{C}\right] - \mu_p^{(k+1)}\right)^2}{Var\left[\hat{\beta}^{(k+1)} - \mathbb{E}\left[\hat{\beta}^{(k+1)}|\mathcal{C}\right]|\mathcal{C}\right]}\right)^{-1} \\
&= \left(1 + \frac{\left(\mathbb{E}\left[\hat{\beta}^{(k+1)}|\mathcal{C}\right] - \mu_p^{(k+1)}\right)^2}{Var\left[\hat{\beta}^{(k+1)}|\mathcal{C}\right]}\right)^{-1} \\
&\leq \left(1 + \frac{\left[\frac{1-\gamma}{\gamma}\mu_p^{(k+1)}\right]^2}{\left(\frac{1}{N+1} + \frac{2}{N+2}\right)(\sigma_c)^2}\right)^{-1}.
\end{aligned}
\tag{36}
$$

Therefore,

$$P\left(\hat{\beta}^{(k+1)} \geq \mu_p^{(k+1)}\Big|\mathcal{C}\right) \geq \frac{\left[\frac{1-\gamma}{\gamma}\mu_p^{(k+1)}\right]^2}{\left[\frac{1-\gamma}{\gamma}\mu_p^{(k+1)}\right]^2 + \left(\frac{1}{N+1} + \frac{2}{N+2}\right)(\sigma_c)^2}. \tag{37}$$

### A.4. Proof for Theorem 4.6

When $k = 0$, we have

$$\mathbb{E}\left[\hat{\beta}^{(1)}\right] = \theta\mu_p^{(0)} \geq \gamma\mu_p^{(0)} \geq \mu_p^{(1)} \tag{38}$$

and

$$Var\left[\hat{\beta}^{(1)}\right] = \frac{1}{N+1}\left(\sigma_p^{(0)}\right)^2 \tag{39}$$

Additionally, we have

$$P\left(\hat{\beta}^{(1)} \leq \mu_p^{(1)}\right) \leq P\left(\hat{\beta}^{(1)} \leq \mu_p^{(0)}\right) = \frac{1}{2^{N+1}}. \tag{40}$$

Then for $k \geq 1$, according to Theorem 4.5,

$$P\left(\hat{\beta}^{(k+1)} \geq \mu_p^{(k+1)}\Big|\mathcal{C}\right) \geq \frac{\left[\frac{1-\gamma}{\gamma}\mu_p^{(k+1)}\right]^2}{\left[\frac{1-\gamma}{\gamma}\mu_p^{(k+1)}\right]^2 + \left(\frac{1}{N+1} + \frac{2}{N+2}\right)(\sigma_c)^2}. \tag{41}$$

Noting the Markov property of $\hat{\beta}^{(k+1)}$ we have

$$P\left(\hat{\beta}^{(k)} \geq \mu_p^{(k)}, 1 \leq k \leq K\right) = P\left(\hat{\beta}^{(1)} \geq \mu_p^{(1)}\right)\prod_{k=1}^{K-1} P\left(\hat{\beta}^{(k+1)} \geq \mu_p^{(k+1)} \mid \hat{\beta}^{(k)} \geq \mu_p^{(k)}\right)$$

$$\geq \left(1 - \frac{1}{2^{N+1}}\right)\prod_{k=1}^{K-1}\left[\frac{\left[\frac{1-\gamma}{\gamma}\mu_p^{(k+1)}\right]^2}{\left[\frac{1-\gamma}{\gamma}\mu_p^{(k+1)}\right]^2 + \left(\frac{1}{N+1} + \frac{2}{N+2}\right)(\sigma_c)^2}\right]. \tag{42}$$

### A.5. Further Discussion on Theorem 4.6

#### A.5.1. NUMERICAL ANALYSIS FOR PROBABILITY BOUND

We conduct a numerical analysis of the probability lower bound presented in Theorem 4.6 for a common scenario. Specifically, we set decent factor $\gamma = 0.9$, quality of large model at the initial step $\mu_p^{(0)} = 0.85$, maximum reasoning quality variance $\sigma_c = 0.01$, drafting size $N = 10$, and maximum reasoning steps $K = 10$. Using Theorem 4.6, we compute the current probability lower bound up to the $k$-th step, $1 \leq k \leq K$. The results of this computation are presented in Figure 5. At the 10-th reasoning step, the probability lower bound remains as high as 0.90. Although this conclusion is derived under highly idealized conditions, it still provides theoretical support for the high quality of thoughts generated from our method.

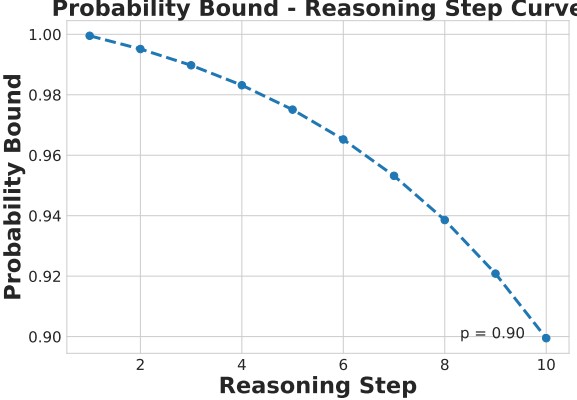

*Figure 5.* The bound decent rapidly with the reasoning steps. However, the bound remains as high as $0.90$ even at the $10$-th step.

### A.5.2. THRESHOLD ESTIMATOR WITH MAXIMUM ESTIMATION

In practice, due to the limited number of samples, the accuracy of the average estimation method tends to be lower, which results in a decrease in the quality of the thoughts generated by our speculative reasoning algorithm. Therefore, we incorporate the solutions from the small model $G_q$ that passed the threshold into the estimation of $\mu_p^{(k)}$ and use the maximum as a non-parametric estimator. Specifically, at reasoning step $k + 1$, denote the set of qualities of thoughts from small model $G_q$ that passed $\tilde{\beta}^{(k)}$ by $\mathcal{V}_q^{(k)} = \left\{ \hat{V}_{i_1}^{(k)}, \hat{V}_{i_2}^{(k)}, \ldots, \hat{V}_{i_{N-M}}^{(k)} \right\}$, and the set of qualities of thoughts generated by large model (speculative model) $G_p$ by $\mathcal{V}_p^{(k)} = \left\{ V_1^{(k)}, V_2^{(k)}, \ldots, V_M^{(k)} \right\}$. Then, our estimator takes the form of:

$$\tilde{\beta}^{(k+1)} = \theta \tilde{\beta}^{(k)} + (1 - \theta) \max \mathcal{V}_p^{(k)} \cup \mathcal{V}_q^{(k)}, \tag{43}$$

with the initial threshold $\tilde{\beta}^{(0)} = \theta \max \mathcal{V}_p^{(0)}$. It's easy to see that $\tilde{\beta}^{(k)} \geq \hat{\beta}^{(0)}, k = 1, 2, \ldots$. Therefore, we have

$$P \left( \tilde{\beta}^{(k+1)} \geq \mu_p^{(k+1)}, 1 \leq k \leq K \right) \geq P \left( \hat{\beta}^{(k+1)} \geq \mu_p^{(k+1)}, 1 \leq k \leq K \right). \tag{44}$$

That indicates that the maximum estimation method results in a higher probability of producing quality-preserved thoughts, at the cost of increased computational resources.

### A.5.3. THRESHOLD ESTIMATOR WITH NO ADDITIONAL $G_p$ SAMPLES

For the sake of simplicity in the previous analysis, we assumed that large model (speculative model) $G_p$ generates $M + 1$ solutions at each step to ensure the existence of the large model's solution. In reality, we can make a more practical assumption that $G_p$ still generates $M$ thoughts and $M \geq 1$. Then $U = N - M$ follows a truncated binomial distribution:

$$P(U = u) = \begin{cases} 0 & u = n, \\ \dbinom{N}{u} \dfrac{p_s^u (1 - p_s)^{N-u}}{1 - p_s^n} & \text{else,} \end{cases} \tag{45}$$

where $p_s = 1 - \Phi \left( \frac{\beta^{(k)} - \mu_q^{(k)}}{\sigma_q^{(k)}} \right)$. We can calculate that

$$\mathbb{E}[U] = \frac{N(p_s - p_s^N)}{1 - p_s^N}. \tag{46}$$

Then

$$\mathbb{E}\left[\overline{V}^{(k)}\right] = \mu_p^{(k)} + \frac{p_s - p_s^N}{1 - p_s^n}(\mu_q' - \mu_p^{(k)}). \tag{47}$$

Therefore we can draw the same conclusion with Theorem 4.3. In addition, we find (32) changes into

$$II_2 = \frac{\left(\sigma_p^{(k)}\right)^2}{N - \mathbb{E}\left[U^{(k)} \mid \mathcal{C}, \hat{\beta}^{(k)}\right]} = \frac{\left(\sigma_p^{(k)}\right)^2}{N - \frac{N(p_s - p_s^N)}{1 - p_s^N}} \leq \frac{2}{N}\left(\sigma_p^{(k)}\right)^2, \tag{48}$$

and the probability bound for quality-preserving changes to

$$P\left(\hat{\beta}^{(k+1)} \geq \mu_p^{(k+1)}, 0 \leq k \leq K\right) \geq \left(1 - \frac{1}{2^N}\right) \prod_{k=0}^{K}\left[\frac{\left[\frac{1-\gamma}{\gamma}\mu_p^{(k+1)}\right]^2}{\left[\frac{1-\gamma}{\gamma}\mu_p^{(k+1)}\right]^2 + \frac{3}{N}\left(\sigma_c\right)^2}\right]. \tag{49}$$

## B. More Background

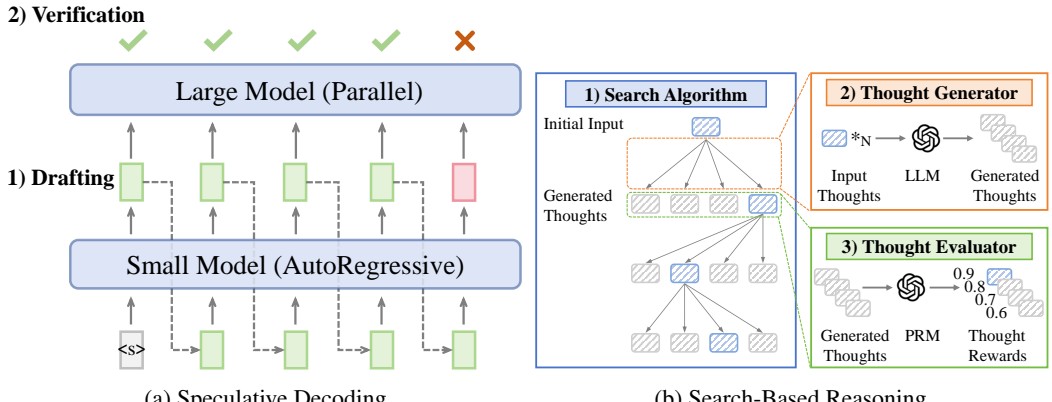

*Figure 6.* (a) Illustration of standard speculative decoding methods. (b) Illustration of the beam-search-based reasoning method.

**Details on Speculative Sampling** Here is a detailed introduction of speculative sampling (SpS) (Leviathan et al., 2023; Chen et al., 2023a), a state-of-the-art decoding technique that significantly accelerates LLM inference while *preserving the target model's distribution*. Specifically, let $c$ denote the prefix, $M_q$ and $M_p$ be the small and large models, respectively, and $\gamma$ represent the number of tokens generated per step. SpS operates in two phases: drafting and verification. In the drafting phase, the small model $M_q$ performs autoregressive sampling to generate $\gamma$ tokens, denoted as $x_1, x_2, \ldots, x_\gamma$, where each $x_i \sim M_q(x_i \mid x_{i-1}, x_{i-2}, \ldots, x_1, c)$. In the verification phase, the large model $M_p$ verifies the tokens generated by $M_q$ in parallel, obtaining the probability distribution $M_p(x \mid x_{i-1}, \ldots, x_1, c)$. Each token $x_i$ is then verified sequentially using a modified rejection sampling mechanism, accepted with probability $\min\left(1, \frac{M_p(x_i \mid x_{i-1}, \ldots, x_1, c)}{M_q(x_i \mid x_{i-1}, \ldots, x_1, c)}\right)$. If $x_i$ is rejected, the verification process terminates, and a resampling phase begins to generate a new token $\tilde{x}_i$. Theoretically, this approach ensures that the distribution of accepted tokens matches that of the large model.

**Details on Beam Search and MCTS** Beam Search is a heuristic search algorithm that starts from the root node and generates $N$ child nodes. At each depth level, only the top $k$ most promising nodes (beam size) are retained. This process is repeated for the selected $k$ nodes until a termination condition is met. The highest-scoring path is returned as the solution. MCTS is a simulation-based decision-making method that starts from the root node and selects child nodes according to a specific strategy (e.g., UCB) until an unexpanded node is reached. A new child node is then expanded. From this new node, a series of random steps are executed to simulate the search process until the end or a predefined depth is reached. After the simulation, rewards propagate back up the tree, updating the value of each visited node based on the simulation results. Through multiple iterations, MCTS converges to an optimal solution.

# C. More Related Work

Here we discuss more related work as follows.

**Reinforcement Learning and Search** Recent advancements in enhancing the reasoning capabilities of large language models (LLMs) broadly fall into two categories: reinforcement learning-based post-training methods (Shao et al., 2024; Yu et al., 2025; Zheng et al., 2024; Hu; Yue et al., 2025) and search-based slow-thinking frameworks (Yao et al., 2023; Gao et al., 2024; Zhang et al., 2024b). Reinforcement learning-based post-training methods closely align with reinforcement learning techniques (Schulman et al., 2015; Zhou et al., 2020; Wang et al., 2023b; Schulman et al., 2017; Wang et al., 2022; Janner et al., 2019; Yang et al., 2022) employed for combinatorial optimization tasks (Berto et al., 2023; Wang et al., 2025; 2024b; Delarue et al., 2020; Wang et al., 2024d; Geng et al., 2024; Mazyavkina et al., 2021; Wang et al., 2023a), predominantly leveraging policy-gradient algorithms to optimize autoregressive models toward improved reasoning performance. In contrast, search-based slow-thinking frameworks augment pretrained LLMs by incorporating external search methodologies—such as beam search (Kang et al., 2024) and Monte Carlo Tree Search (MCTS) (Gao et al., 2024; Liu et al., 2023; Zhang et al., 2024b; Guo et al., 2024)—to enable systematic and structured deliberation without necessitating further training. This paper specifically emphasizes the exploration and advancement of these search-based slow-thinking frameworks.

# D. Implementation Details of the Baselines

We implement the baselines used in our paper based on the OpenR code framework.

## D.1. AR

AR refers to Autoregressive Generation, a sequence-based model generation method widely used in language models. In the standard Tree of Thoughts (ToT) method, autoregressive generation involves constructing solutions step by step, where each step uses information from previous steps to guide current choices. This approach is straightforward and intuitive but can be limited in terms of speed. In our work, the AR methods using Beam Search and MCTS are based on existing open-source code from OpenR.

## D.2. SpS

SpS refers to Speculative Sampling, a parallel decoding method for model generation. By introducing a small model, speculative sampling accelerates the generation process in tree search reasoning methods. We implement an efficient SpS method for Beam Search and MCTS using the vLLM (Kwon et al., 2023) package, building on the open-source code from OpenR.

# E. Details of the Datasets Used in This Paper

## E.1. The Datasets Used in the Main Evaluation

**MATH-100** The MATH dataset (Hendrycks et al., 2021) consists of 12,500 challenging competition mathematics problems, each with a full step-by-step solution. These solutions can be used to teach models to generate answer derivations and explanations. We randomly select 100 problems from this dataset as our test set. We choose this number of problems for our test set because the inference latency of the tree search algorithm is quite long. Even with our efficient SpecSearch acceleration framework, the latency remains significant. To avoid the experiment running time being too long for a single run, we select 100 problems for evaluation.

**GSM8K-100** The GSM8K dataset (Cobbe et al., 2021) contains 8.5K high-quality, linguistically diverse grade school math word problems. We randomly select 100 problems from this dataset for our test set. The reason for selecting 100 problems is the same as for the MATH dataset: to manage inference latency effectively.

## E.2. The Datasets Used in the Ablation Study

**MATH-50** For the ablation study, we need to test multiple variants of SpecSearch and conduct hyperparameter robustness experiments, which requires running the experiments multiple times. To facilitate this process, we select 50 mathematical problems from the MATH dataset as the test set for the ablation study.

## F. Illustration of Using Models

**Thought Generator** For the Qwen series of models, we use the Qwen2.5-72B-Instruct-GPTQ-Int4 model as the large model and the Qwen2.5-7B-Instruct-GPTQ-Int4 model as the small model. For the Llama series of models, we use the Llama-3-70B-Instruct-GPTQ-Int4 model as the large model and the Llama-3-8B-Instruct-GPTQ-Int4 model as the small model.

**Thought Evaluator** We use two PRM models as thought evaluators, one is Math-Shepherd (Wang et al., 2024c) and the other is Math-psa (Wang et al., 2024a) for **Experiment 2**.

## G. Discussion on the novelty of SpecSearch over standard speculative decoding and TreeBon (Qiu et al., 2024)

### G.1. Comparison with Existing Speculative Decoding Techniques

**Relation to Standard Speculative Decoding (SD) Methods.** We discuss the novelty of *SpecSearch* compared to existing SD techniques, emphasizing **key distinctions** in terms of **speculative formulation**, **verification and rejection strategies**, and **theoretical guarantees**.

- **Bi-Level Speculative Formulation:** Unlike existing SD methods focused solely on tokens, *SpecSearch* treats both high-level thoughts and low-level tokens as bi-level speculative tasks. This enables (1) **Structural Alignment** with reasoning frameworks, where thoughts are fundamental units, and (2) **Compatibility** with standard SD methods through low-level token-level speculation.

- **Contextual Verification for Higher Acceptance and Speedup:** Unlike SD methods that enforce strict token-level alignment, leading to frequent rejections, *SpecSearch* verifies the **contextual quality** of reasoning thoughts. This allows acceptance of correct but non-aligned outputs, substantially boosting acceptance rates and achieving significant speedups.

- **Quality-Preserving Rejection Mechanism:** In contrast to token-level rejection in standard SD methods, *SpecSearch* introduces **quality-preserving thought-level rejection** based on contextual quality. It discards entire thoughts only when their quality is lower than the large model's, ensuring high-quality reasoning throughout decoding.

- **Theoretical Guarantee of Reasoning Quality:** While standard SD methods preserve token-level distributions, *SpecSearch* guarantees that the reasoning quality remains comparable to that of the large model.

### G.2. Comparison with Treebon (Qiu et al., 2024)

We discuss the novelty of *SpecSearch* compared to Treebon (Qiu et al., 2024), emphasizing key distinctions in terms of **motivation**, **speculative formulation**, **rejection strategies**, and **theoretical guarantees**.

- **Distinct Motivation:** Unlike Treebon, which aims to accelerate best-of-$n$ sampling via speculative rejection and tree search, *SpecSearch* is the first to **generalize speculative execution to LLM reasoning tasks**.

- **Bi-Level Speculative Formulation:** Treebon treats fixed-length token sequences as speculative units, while *SpecSearch* adopts a **flexible bi-level formulation**—modeling full reasoning thoughts as high-level tasks and tokens as low-level ones. Unlike Treebon's fixed-length design, *SpecSearch* leverages LLMs' reasoning capabilities to generate semantically coherent thoughts of dynamic length.

- **Quality-Preserving Rejection Mechanism:** Treebon rejects a fixed proportion of token sequences using a preset threshold. In contrast, *SpecSearch* scores reasoning thoughts and **adaptively rejects those with lower contextual quality relative to the large model's output**, enabling finer control and better quality preservation.

- **Theoretical Guarantee:** Unlike Treebon, which lacks theoretical guarantees, *SpecSearch* offers **formal assurance** that the quality of the output reasoning remains on par with that of the large model.

---

**Algorithm 2** Pseudo Code for SpecSearch

---

**Input:** Input question $c$, large model (speculative model) $G_p$, small model $G_q$, evaluation model $V$, expansion width $N$, beam size $b$, EMA weight $\theta$, reasoning depth $K$, a nonparametric estimation method $\Theta$.

Initialize beam: $\mathcal{B} \leftarrow \emptyset$              ▷ Each element takes the form of $[\text{sequence}, \text{quality}]$

Initialize candidate thoughts: $\mathcal{T} \leftarrow \emptyset, \mathcal{V} \leftarrow \emptyset$           ▷ Initial reasoning from large model $G_p$

**for** $i = 1$ **to** $N$ **do**

    Generate from large model: $z \leftarrow G_p(\cdot \mid c)$

    Evaluate generated thought: $v \leftarrow V(z)$

    Update candidates: $\mathcal{T} \leftarrow \mathcal{T} \cup \{(z, c)\}, \mathcal{V} \leftarrow \mathcal{V} \cup \{v\}$

**end for**

Initialize threshold: $\hat{\beta}^{(1)} \leftarrow \theta\Theta(\mathcal{V})$

Update beam: $\mathcal{B} \leftarrow \text{Top}_b(\mathcal{T})$                  ▷ Retain top $b$ by quality

**for** $k = 1$ **to** $K$ **do**

    Initialize candidate thoughts: $\mathcal{T} \leftarrow \emptyset$

    **for** $z_{<k}$ **in** $\mathcal{B}$ **do**

        Generate Thoughts: $\hat{\beta}^{(k+1)}, \mathcal{T}_i \leftarrow G_s\left(z_{<k}, G_p, G_q, V, \hat{\beta}^{(k)}, N, \theta, \Theta\right)$      ▷ Speculatively search

        Update candidate thoughts: $\mathcal{T} \leftarrow \mathcal{T} \cup \mathcal{T}_i$

    **end for**

    Update beam: $\mathcal{B} \leftarrow \text{Top}_b(\mathcal{T})$              ▷ Retain top $b$ by quality

    **if** $\forall z_{\leq k} \in \mathcal{B}, \text{last}(z_{\leq k}) = \texttt{<STOP>}$ **then**

        **break**                      ▷ Finish searching

    **end if**

**end for**

**return** $\mathcal{B}$

---

## H. Implementation Details of Our SpecSearch

### H.1. Discussion on Advantages of Our Evaluation Method

Here we present a detailed discussion on using a process reward model to evaluate the quality of thoughts. **First**, the thought evaluator accurately captures a thought's complete semantic meaning. **Second**, it converts thought distribution into a structured, manageable quality distribution, enabling a clearer definition of lossless reasoning acceleration. **Third**, it assigns high scores to different valid reasoning paths, improving the assessment of the small model's thought quality.

### H.2. SpecSearch Implementation Details

#### H.2.1. SMALL MODEL PARALLEL THOUGHT GENERATION

Due to the small memory footprint of small models, they can operate in parallel even under limited memory conditions. Generating multiple thoughts simultaneously does not significantly increase latency compared to generating a single thought. Therefore, we use a small model to generate thoughts in parallel. Although the overall quality of generation from small models may not match that of large models, they still produce high-quality thoughts.

We utilize the small model to generate 2*N thoughts in parallel, combining the efficiency of parallel processing with the ability to generate high-quality thoughts. This approach introduces more high-quality thoughts into the Tree of Thoughts (ToT), enhancing both efficiency and thought quality.

#### H.2.2. ACCEPTANCE-REJECTION MECHANISM

After generating 2*N thoughts with the small model, we evaluate these thoughts using the Process Reward Model (PRM) to determine their rewards. Each thought's reward is compared to a dynamically calculated threshold. If the reward surpasses the threshold, the thought is retained; otherwise, it is discarded. If more than N thoughts are retained, we select the top N thoughts with the highest rewards for final acceptance.

#### H.2.3. ALGORITHM IMPLEMENTATION

*Table 4.* **Full GSM8K.** Evaluation on the full GSM8K-1319 dataset. **(1) Setup** We utilize quantized versions of Qwen2.5-72B-Instruct and Qwen2.5-7B-Instruct as the large and small language models, utilize MATH-psa as the Process Reward Model and employ beam search as the search algorithm. Unless explicitly stated otherwise, all results presented below follow this setting. **(2) Results** The results demonstrate that SpecSearch achieves comparable accuracy while significantly reducing inference latency.

| | Qwen models | | | |
| --- | --- | --- | --- | --- |
| MATH Dataset | GSM8K-1319 | | | |
| Methods | Reasoning Accuracy (%) | Average Inference Latency (s) | Speedup (vs AR) | Speedup (vs SpS) |
| AR | 96.66 | 144.63 | NA | 0.48 |
| SpS | 96.66 | 70.04 | 2.06 | NA |
| SpecSearch (Ours) | 95.83 | 50.99 | 2.84 | 1.37 |

*Table 5.* **AIME.** Evaluation on the AIME dataset. The results demonstrate that SpecSearch achieves comparable accuracy while significantly reducing inference latency.

| | Qwen models | | | |
| --- | --- | --- | --- | --- |
| MATH Dataset | AIME | | | |
| Methods | Reasoning Accuracy (%) | Average Inference Latency (s) | Speedup (vs AR) | Speedup (vs SpS) |
| AR | 16.67 | 562.89 | NA | 0.57 |
| SpS | 13.33 | 318.71 | 1.77 | NA |
| SpecSearch (Ours) | 13.33 | 264.44 | 2.13 | 1.21 |

The procedure of our bi-level speculative thought generator is outlined in Algorithm 1 in the main text. Here, we further present the complete SpecSearch algorithm, which is based on the beam search algorithm, as shown in Algorithm 2.

Furthermore, due to the limited sample size $M$, we adopt a more conservative estimation strategy in the implementation, utilizing the maximum value as an estimate of the upper confidence bound for $\mu_p^{(k+1)}$. Specifically, let $\mathcal{V}_q^{(k)}$ denote the set of qualities of thoughts generated by the small model $G_q$, and $\mathcal{V}_p^{(k)}$ denote the set of qualities of thoughts generated by the large model (speculative model) $G_p$. The threshold estimation method we employ is as follows:

$$\tilde{\beta}^{(k+1)} = \theta\tilde{\beta}^{(k)} + (1 - \theta)\max \mathcal{V}_p^{(k)} \cup \mathcal{V}_q^{(k)}. \tag{50}$$

### H.2.4. SPECULATIVE MODEL SERIAL THOUGHT GENERATION

If the number of accepted thoughts from the small model is less than N after filtering through the acceptance-rejection mechanism, we use a speculative model to serially generate additional thoughts until the total number of thoughts reaches N.

### H.3. Hyperparameters

**SpecSearch** In our experiments, unless otherwise specified, we set the EMA weight $\theta$ in the SpecSearch to 0.9.

**Beam Search** In our experiments, unless otherwise specified, we set the tree width to 6, the tree depth to 50, and the beam size to 2 in the Beam Search.

**MCTS** In our experiments, unless otherwise specified, we set the tree width to 6, the tree depth to 50, and the iteration number to 4 in the MCTS.

## I. More Results

### I.1. More Main Evaluation

We conduct comprehensive evaluations across **three distinct dataset categories** to rigorously demonstrate the efficiency and generalizability of *SpecSearch*. Specifically, these include: (1) the **full GSM8K** dataset comprising 1,319 problems; (2) more challenging mathematical reasoning benchmarks, namely the **AIME** and **Olympiad** datasets; and (3) a **code-generation** benchmark. As illustrated in Tables 4, 5, 6, and 7, *SpecSearch* **consistently** and **significantly surpasses state-of-the-art approaches** across all three dataset categories, achieving speedups ranging from **2.04×** to **2.84×** while maintaining

*Table 6.* **Olympiad Bench.** Evaluation on the Olympiad Bench (OE-TO-maths-zh-CEE) dataset. The results demonstrate that SpecSearch achieves comparable accuracy while significantly reducing inference latency.

| | Qwen models | | | |
|---|---|---|---|---|
| MATH Dataset | Olympiad Bench | | | |
| Methods | Reasoning Accuracy (%) | Average Inference Latency (s) | Speedup (vs AR) | Speedup (vs SpS) |
| AR | 63.75 | 358.44 | NA | 0.67 |
| SpS | 58.75 | 241.80 | 1.48 | NA |
| SpecSearch (Ours) | 58.75 | 176.02 | 2.04 | 1.37 |

*Table 7.* **Code-Generation Benchmark.** Evaluation on the HumanEval dataset. The results show that SpecSearch achieves comparable accuracy while significantly reducing inference latency.

| | Qwen models | | | |
|---|---|---|---|---|
| Coding Dataset | HumanEval | | | |
| Methods | Reasoning Accuracy (%) | Average Inference Latency (s) | Speedup (vs AR) | Speedup (vs SpS) |
| AR | 85.37 | 342.18 | NA | 0.65 |
| SpS | 84.15 | 223.30 | 1.53 | NA |
| SpecSearch (Ours) | 85.37 | 158.43 | 2.16 | 1.41 |

comparable reasoning accuracy. These findings highlight *SpecSearch*'s versatility and robustness, demonstrating substantial improvements in inference speed with minimal or no compromise in accuracy **across diverse tasks**.

**Setup.** Throughout our experiments, we utilize quantized versions of `Qwen2.5-72B-Instruct` and `Qwen2.5-7B-Instruct` as the large and small language models, respectively. Additionally, we incorporate `MATH-psa` as the Process Reward Model and employ beam search as the search algorithm.

**Results.**

**(1) Full GSM8K Dataset (1,319 Problems):** *SpecSearch* achieves a substantial **2.84× speedup** compared to the AR baseline, with only a minimal accuracy reduction of 0.83%. This result highlights *SpecSearch*'s capability to effectively scale to larger problem sets while preserving high reasoning accuracy.

**(2) High-Difficulty Mathematics (AIME and Olympiad Bench):** We conduct experiments on the AIME and Olympiad Bench (`OE_TO_maths-zh_CEE`) datasets. Notably, *SpecSearch* **maintains identical accuracy** to the SpS method while achieving **speedups of 1.21× and 1.37×**, respectively. These results demonstrate the method's effectiveness in handling challenging, competition-level mathematics problems.

**(3) Code Generation (HumanEval):** To assess *SpecSearch* beyond mathematical reasoning, we evaluate its performance on the HumanEval code-generation benchmark. The results show that *SpecSearch* achieves a **2.16× speedup** over the AR without any reduction in accuracy. Furthermore, it **surpasses the SpS by 1.22% in accuracy** while simultaneously delivering a **1.41× speedup**. These results underscore *SpecSearch*'s strong generalization capabilities across diverse domains.

### I.2. More Motivating Results

**Reward Distribution Across Reasoning Steps** We analyze the reward distributions across different reasoning steps in our experiments. Figure 7 shows the reward distribution for each step in the reasoning path. The figure illustrates that the average reward decreases as the reasoning process moves from initial steps to later stages.

Initially, reasoning steps tend to yield higher rewards because they are simpler and require less cognitive effort, allowing for higher thresholds. As the reasoning progresses, subsequent steps become more complex, resulting in lower average reward scores and necessitating lower thresholds. This pattern supports our approach of using dynamic thresholds.

**Similar Output Lengths Enable Effective Model Collaboration** We calculate the average number of tokens generated by large and small models in a single reasoning step. Table 8 shows that both model types produce a similar number of tokens. This similarity suggests that the length of the thought or reasoning process at each step is comparable. This feature supports

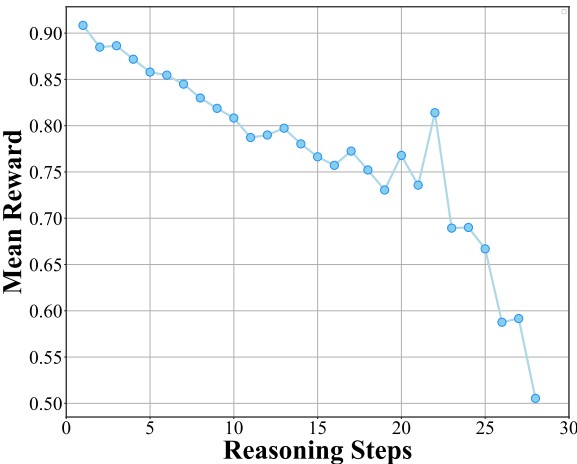

*Figure 7.* The distribution of rewards for generated thoughts decreases step by step.

*Table 8.* The results demonstrate that the number of tokens generated by large and small models in a single reasoning step is comparable.

| Model | The average number of tokens in a reasoning step |
|---|---|
| Qwen2.5-7B-Instruct | 59.088 |
| Qwen2.5-1.5B-Instruct | 57.037 |

collaboration between large and small models, as it implies they can efficiently divide tasks and work together within the same reasoning framework.

### I.3. Case Study

**Case 1** Figure 8 shows a challenging case study. It illustrates how different reasoning steps vary in difficulty. In this process, identifying the curve type from its equation is an easy step, while transforming an equation from polar to Cartesian coordinates is more difficult.

**Case 2** Figure 9 presents a simple case study showing how different reasoning steps have different levels of difficulty. In this reasoning process, calculating $9900 + 1$ is an easy step while computing the square of 99 is a hard step.

Figures 8 and 9 illustrate varying difficulty levels among reasoning steps. Simpler steps are efficiently handled by a small model, while more complex steps are managed by a large model. This division optimizes efficiency and accuracy throughout the reasoning process.

**Case 3** We select a problem from the GSM8K-100 dataset where SpecSearch made an error for case study analysis. Figure 10 shows the PRM score on the incorrect reasoning path. In this scenario, the first three reasoning steps are correct, but an error occurs at the fourth step. Subsequent steps remain incorrect. Notably, the fourth step, despite being wrong, achieves a high PRM score of 0.8916015625. This indicates that incorrect steps can mislead the PRM and prevent it from accurately identifying errors. This observation clarifies why we observed low precision loss in our SpecSearch.

### I.4. More Broad Compatibility Results

In this section, we provide more results of the broad compatibility experiment. We conduct broad compatibility experiments on the MATH-100 dataset using the Qwen models. The results in Table 9 show the performance of SpecSearch and the baselines in different search algorithms and different thought evaluators. SpecSearch accelerates beam search and MCTS, outperforming baselines by reducing latency with minimal accuracy loss, and shows consistent performance across different PRMs, demonstrating broad applicability and generalization.

This experimental result supplements the broad compatibility experiment in the main text, verifying that our method has broad applicability across different datasets.

*Table 9.* The results demonstrate the Broad Compatibility of Our SpecSearch with different search algorithms and PRMs on the MATH-100 dataset.

| Search Algorithms | Beam Search | | | | MCTS | | | |
|---|---|---|---|---|---|---|---|---|
| Methods | Reasoning Accuracy (%) ↑ | Average Inference Latency (s) ↓ | Speedup (vs AR)↑ | Speedup (vs SpS)↑ | Reasoning Accuracy (%) ↑ | Average Inference Latency (s) ↓ | Speedup (vs AR) ↑ | Speedup (vs SpS) ↑ |
| AR | 87.00 | 275.78 | NA | 0.51 | 93.00 | 523.54 | NA | 0.49 |
| SpS | 88.00 | 141.55 | 1.95 | NA | 91.00 | 257.62 | 2.03 | NA |
| SpecSearch (Ours) | 87.00 | **82.35** | **3.35** | **1.72** | 90.00 | **171.59** | **3.05** | **1.50** |

| PRMs | Math-psa | | | | Math-Shepherd | | | |
|---|---|---|---|---|---|---|---|---|
| Methods | Reasoning Accuracy (%) ↑ | Average Inference Latency (s) ↓ | Speedup (vs AR) ↑ | Speedup (vs SpS) ↑ | Reasoning Accuracy (%) ↑ | Average Inference Latency (s) ↓ | Speedup (vs AR) ↑ | Speedup (vs SpS) ↑ |
| AR | 87.00 | 275.78 | NA | 0.51 | 88.00 | 265.29 | NA | 0.55 |
| SpS | 88.00 | 141.55 | 1.95 | NA | 85.00 | 145.53 | 1.82 | NA |
| SpecSearch (Ours) | 87.00 | **82.35** | **3.35** | **1.72** | 85.00 | **118.67** | **2.24** | **1.23** |

*Table 10.* **Sensitivity to Draft Models.** We investigate SpecSearch's performance using multiple small draft models—Qwen2.5-3B-Instruct, Qwen2.5-1.5B-Instruct, and Qwen2.5-0.5B-Instruct. The results demonstrate that our method maintains stable accuracy while achieving significant latency reduction across various draft models.

| MATH Dataset | GSM8K-100 | | | | |
|---|---|---|---|---|---|
| Methods | Reasoning Accuracy (%) | Average Inference Latency (s) | Speedup (vs AR) | Speedup (vs SpS) | Draft Acceptance Rate (%) |
| AR | 97 | 138.24 | NA | 0.50 | NA |
| SpS (Draft-7B) | 97 | 69.43 | 1.99 | NA | NA |
| SpecSearch (Ours, Draft-7B) | 96 | 48.18 | 2.87 | 1.44 | 49.19 |
| SpecSearch (Ours, Draft-3B) | 96 | 63.48 | 2.18 | 1.09 | 44.54 |
| SpecSearch (Ours, Draft-1.5B) | 95 | 53.49 | 2.58 | 1.30 | 45.79 |
| SpecSearch (Ours, Draft-0.5B) | 96 | 49.54 | 2.79 | 1.40 | 35.48 |

## I.5. Sensitivity Analysis to Draft Model's Size

We have investigated *SpecSearch*'s performance using multiple small draft models. The results in Table 10 reveal that *SpecSearch* achieves **speedups ranging from 2.18× to 2.87×**, underscoring its **robust acceleration capabilities across diverse small-model settings**.

## I.6. More Ablation Study Results

**Sensitivity Analysis** Hyperparameter $\theta$, which controls the relative importance of reward information from the previous layer when updating the threshold for the current layer, is crucial for balancing between accuracy and latency. To understand the impact of hyperparameter $\theta$ on the performance of SpecSearch, we conduct a detailed sensitivity analysis focusing exclusively on this parameter.

We vary $\theta$ across a range from $\theta_{\min} = 0.8$ to $\theta_{\max} = 0.95$, with increments of $\Delta\theta = 0.05$. For each value of $\theta$, we evaluate SpecSearch using the MATH-50 dataset, ensuring that all other hyperparameters are held constant to isolate the effect of $\theta$.

The results in Table 11 show that the accuracy of SpecSearch remains largely unchanged when $\theta$ is large and latency decreases as $\theta$ increases. These findings suggest that while $\theta$ does not significantly affect accuracy, setting $\theta$ closer to 1 can lead to substantial improvements in computational efficiency without compromising the quality of the generated reasoning paths. This demonstrates the robustness of $\theta$.

*Table 11.* The results demonstrate that SpecSearch achieves similar average performance across a wide range of $\theta$.

| Dataset | MATH-50 | | |
|---|---|---|---|
| Methods | Reasoning Accuracy (%) ↑ | Average Inference Latency (s)↓ | Speedup (vs AR)↑ |
| AR | 88.00 | 256.05 | NA |
| SD | 90.00 | 132.68 | 1.93 |
| SpecSearch (Ours,$\theta = 0.95$) | 88.00 | 66.48 | 3.85 |
| SpecSearch (Ours,$\theta = 0.9$) | 88.00 | 70.63 | 3.63 |
| SpecSearch (Ours,$\theta = 0.85$) | 88.00 | 73.66 | 3.48 |
| SpecSearch (Ours,$\theta = 0.8$) | 88.00 | 83.29 | 3.07 |

**Q&A Case 1**

**Question:**

Find the curve defined by the equation $r^2 \cos 2\theta = 4$. Which of the following does this represent?
(A) Line    (B) Circle    (C) Parabola    (D) Ellipse    (E) Hyperbola.
Enter the letter of the correct option.

**Reason:**

**Step 1**: Recall the polar to Cartesian coordinate transformations:

$$x = r \cos \theta, y = r \sin \theta$$

and the identity:
$$\cos 2\theta = \cos^2 \theta - \sin^2 \theta$$

**Step 2**: Substitute $r \cos \theta$ and $r \sin \theta$ into the identity:    → hard

$$\cos 2\theta = \left(\frac{x}{r}\right)^2 - \left(\frac{y}{r}\right)^2 = \frac{x^2}{r^2} - \frac{y^2}{r^2} = \frac{x^2 - y^2}{r^2}$$

**Step 3**: Substitute $\cos 2\theta$ back into the original equation:

$$r^2 \cos 2\theta = 4 \implies r^2 \left(\frac{x^2 - y^2}{r^2}\right) = 4 \implies x^2 - y^2 = 4$$

**Step 4**: Analyze the resulting Cartesian equation:    → easy

$$x^2 - y^2 = 4$$

This is the standard form of a hyperbola. Specifically, it is a hyperbola centered at the origin with its transverse axis along the x-axis.

So the answer is:
$$\boxed{E}$$

*Figure 8.* A difficult case study showing how different reasoning steps have different levels of difficulty.

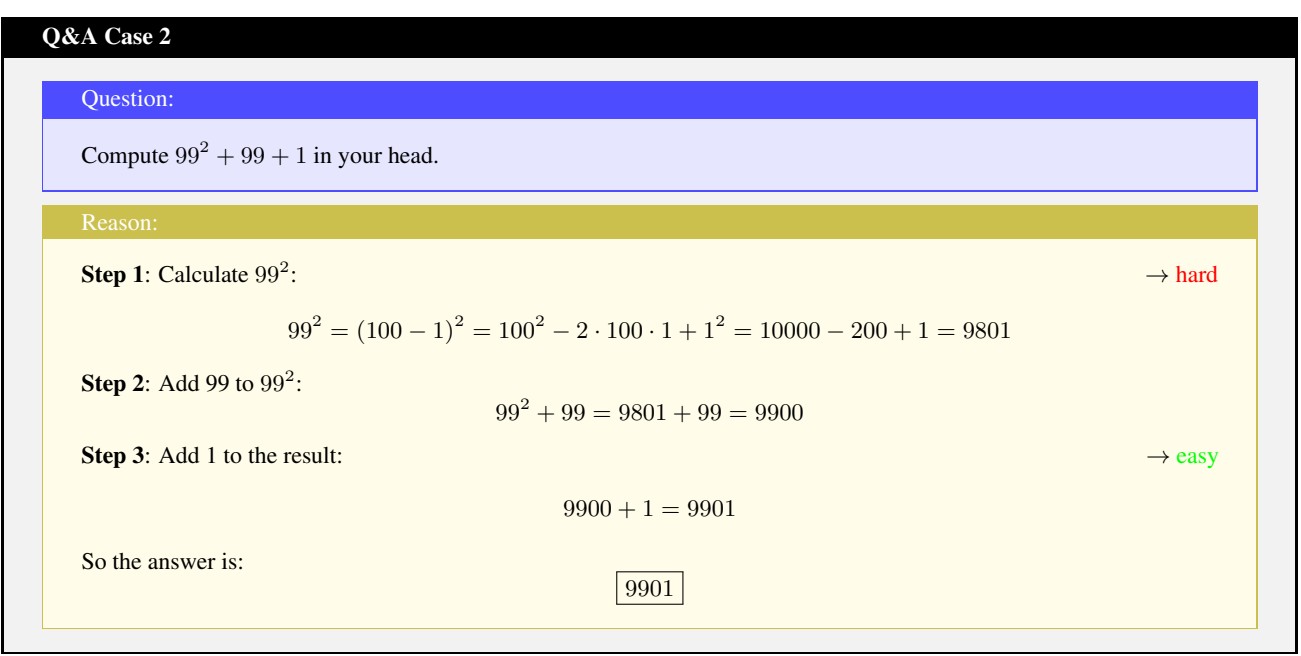

*Figure 9.* A simple case study showing how different reasoning steps have different levels of difficulty.

**Q&A Case 3**

**Question:**

Raymond and Samantha are cousins. Raymond was born 6 years before Samantha. Raymond had a son at the age of 23. If Samantha is now 31, how many years ago was Raymond's son born?

**Reason:**

**Step 1**: Identify the current age of Samantha: Samantha is currently 31 years old. [0.93701171875]

**Step 2**: Determine Raymond's age when his son was born: Raymond had a son at the age of 23. [0.91845703125]

**Step 3**: Find the age difference between Raymond and Samantha: We are told that Raymond was born 6 years before Samantha. This means Samantha is 6 years younger than Raymond. [0.92529296875]

**Step 4**: Calculate Raymond's current age: If Samantha is 31 and she is 6 years older than Raymond, then Raymond is currently 31 - 6 = 25 years old. [0.8916015625]

**Step 5**: Determine the age of Raymond when his son was born: Raymond had his son at the age of 23. [0.81982421875]

**Step 6**: Calculate the number of years since Raymond's son was born: Since Raymond is currently 25 years old and he had his son at age 23, it has been $25 - 23 = 2$ years since Raymond's son was born. [0.56982421875]

So the answer is:

$\boxed{2}$

*Figure 10.* A case study showing the PRM score on the wrong reasoning path.

