# OpenReview forum: "Accelerating Large Language Model Reasoning via Speculative Search"
_ICML.cc/2025/Conference — ICML 2025 poster_

### Official Review · Reviewer_zUgA · 2025-02-14

**Overall Recommendation:** 3

**Summary:**

**I am not very familiar with the subarea of this paper, so I am not highly confident in my review. I expect my review not to play a significant role in the decision.**

This paper introduces SpecSearch, a method that enables a small model to collaborate with a large model at both the thought and token levels in reasoning. It accelerates TSB methods using a large LLM.

## update after rebuttal
Again, **I am NOT familiar with the subarea of this paper, so I expect my review NOT to play a significant role in the decision.**

**Claims And Evidence:**

N/A

**Essential References Not Discussed:**

N/A

**Experimental Designs Or Analyses:**

N/A

**Methods And Evaluation Criteria:**

N/A

**Other Comments Or Suggestions:**

I feel that the writing could be improved. It was difficult for me to understand the framework clearly, and I do feel that the writing makes something overly complicated.

**Other Strengths And Weaknesses:**

N/A

**Questions For Authors:**

N/A

**Relation To Broader Scientific Literature:**

N/A

**Theoretical Claims:**

N/A

---

> ### Author Rebuttal · Authors · 2025-04-01
>
> # Response to Reviewer zUgA
>
> We sincerely thank the reviewer for the valuable comments! We address the concerns in detail as follows. We sincerely hope that our response could properly address your concerns. If so, we would deeply appreciate it if you could raise your score. If not, please let us know your further concerns, and we will continue actively responding to your comments and improving our submission.
>
> ### Weakness 1.
> > **1. I feel that the writing could be improved. It was difficult for me to understand the framework clearly, and I do feel that the writing makes something overly complicated.**
>
> Thank you for the valuable suggestion. We provide a clearer description of our framework below and will incorporate it into the revised version of our paper.
>
> - **Search Framework Overview**: Our framework represents **tree nodes** as intermediate reasoning steps (thoughts) and **tree paths** as candidate solutions to multi-step reasoning problems. Each reasoning step comprises a sequence of tokens decoded by a large language model (LLM). The framework consists of three core components: a **Bi-Level Speculative Thought Generator (G)**, a **Thought Evaluator (V)**, and a **Search Algorithm**. Starting from the root node (containing the input context and question), the generator G expands the reasoning tree by producing $N$ candidate thoughts (child nodes). The evaluator V assesses their quality, which then guides the search algorithm. This iterative process builds a reasoning tree, ultimately selecting a final reasoning path.
>
> - **Bi-Level Speculative Thought Generator**: At each leaf node, the bi-level speculative thought generator produces $N$ high-quality child thoughts **efficiently**, leveraging a **draft-evaluate-reject-correct** paradigm. This process operates on two levels as follows.
>
>   (1) **Drafting (Thought-Level)**: A small model rapidly drafts multiple candidate reasoning thoughts.
>
>   (2) **Evaluating (Thought-Level)**: These drafts are scored by the thought evaluator to assess their contextual quality.
>
>   (3) **Rejection (Thought-Level)**: Thoughts deemed lower in quality than the large model’s outputs are discarded.
>
>   (4) **Correction (Token-Level)**: Rejected thoughts are refined using a lossless token-level speculative decoding method, ensuring accuracy and robustness.
>
> This bi-level design allows SpecSearch to **accelerate generation** significantly while maintaining **high-quality reasoning** throughout the search process.

---

> > ### Comment · Reviewer_zUgA · 2025-04-01
> >
> > Thanks for your clarification. I think the framework is clearer to me now. It would be great to illustrate the high-level picture of your framework in the paper at the very beginning (of the methodology section)!
> >
> > However, as I said in the review, "**I am not very familiar with the subarea of this paper, so I am not highly confident in my review. I expect my review not to play a significant role in the decision.**" So I still don't think I am able to give reliable judgment. Sorry about that.

---

> > > ### Author Response · Authors · 2025-04-02
> > >
> > > Dear Reviewer zUgA,
> > >
> > > Thank you very much for taking the time to review our paper and provide valuable comments and suggestions. We truly appreciate your effort, especially given your note about not being deeply familiar with this subarea.
> > >
> > > We are glad that the clarification helped make the framework clearer. Following your suggestion, we will include a high-level illustration at the beginning of the methodology section to better convey the core ideas of our framework to the readers.
> > >
> > > Once again, thank you for your thoughtful feedback—it has helped us improve the quality and clarity of the paper.

---

### Official Review · Reviewer_EXez · 2025-03-14

**Overall Recommendation:** 4

**Summary:**

This paper introduces a new LLM reasoning method via speculative Tree-Search-Base reasoning. It involved a quality-preserving rejection mechanism and also has theoretical properties that means it can maintain reasoning quality compared to the large model. Experiments on math problems show up to 2–3× speedup over standard autoregressive decoding or token-level speculative decoding, with minimal accuracy drop.

## update after rebuttal

I thank the authors for their detailed rebuttal. I maintain my score after reading all the other rebuttal comments.

**Claims And Evidence:**

* "A Novel SpecSearch Framework". This claim is substantiated.
* "Quality-Preserving Rejection Mechanism". This claim is substantiated.
* "Theoretical Guarantee", also substantiated.
* "Significant Speedup and Versatility" - Empirical evidence is only given with two mathematical reasoning datasets of MATH and GSM8K, I encourage the authors to also use more, perhaps non-mathematical reasoning datasets if possible.

**Essential References Not Discussed:**

All essential references are discussed.

**Experimental Designs Or Analyses:**

Experiments are valid, and use MATH and GSM8K math reasoning benchmarks. A potential issue with the current setup is that the just focusing on these two math datasets could mean the approach performs of an unknown ability to other reasoning domains and or diverse tasks.

**Methods And Evaluation Criteria:**

The method appears to be novel, and well structured, using a small LM for draft thoughts, and a Large LM for final verification, coupled with a tree-based or beam-based search search and process reward model for scoring. The chosen evaluation metrics of accuracy and inference latency are appropriate, however the results can all benefit from error bars from running across random seeds, and more baselines.

**Other Comments Or Suggestions:**

* Compare wall-clock time on real hardware with parallelization overhead considered.
* Expand the discussion on failure cases where the small model’s generation misleads the system.

**Other Strengths And Weaknesses:**

Significance: Significant inference speedup without heavily compromising accuracy with the proposed approach. However the paper could benefit from comparisons with newer speculative approaches.
Clarity: The paper is well-written and easy to follow.
Originality: This approach appears to be novel.

**Questions For Authors:**

* How sensitive is speedup to the small draft model’s performance?
* Can the authors evaluate on a dataset that is not within the math domain?

**Relation To Broader Scientific Literature:**

Speculative decoding and chain-of-thought search approaches (like Tree-of-Thoughts, SEED) are cited, but:
* The paper would benefit from a deeper comparison with closely related works like SEED or other structured speculative decoding methods.

**Theoretical Claims:**

The paper provides probabilistic bounds that, with sufficient sampling and proper thresholds, SpecSearch retains or approximates the large model’s accuracy. I did not check the proofs in detail, intuitively they make sense.

---

> ### Author Rebuttal · Authors · 2025-04-01
>
> # Response to Reviewer EXez
>
> We sincerely thank the reviewer for the thoughtful and encouraging feedback! We hope our response has addressed your concerns. If so, we would be truly grateful if you would consider raising your score. If not, we welcome any further comments and will continue working diligently to improve our work.
>
> > ## Due to limited space, **we provide Tables and Figures in 14604.pdf in [Anonymous Link](https://anonymous.4open.science/r/SpecSearch-Rebuttal-4B78/14604.pdf)**.
>
> ### Weakness 1.
>
> > **1. Use more, non-mathematical reasoning datasets**
>
> We have conducted comprehensive evaluations across **three distinct dataset categories** to rigorously demonstrate the efficiency and generalizability of SpecSearch. Specifically, these include: (1) the **full GSM8K** dataset comprising 1,319 problems; (2) more challenging mathematical reasoning benchmarks, namely the **AIME** and **Olympiad** datasets; and (3) a **code-generation** benchmark. The results show that SpecSearch **consistently** and **significantly surpasses state-of-the-art approaches** across all three dataset categories, achieving speedups ranging from **2.04$\times$** to **2.84$\times$** while maintaining comparable reasoning accuracy. Please refer to **Weakness 1** in **Response to Reviewer cNpL** for detailed results.
>
> ### Weakness 2.
> > **2. Error bars from running across random seeds**
>
> We have evaluated SpecSearch and baselines under three random seeds to assess stability and performance. As shown in Figure 1 in 14604.pdf, SpecSearch delivers **consistent reasoning accuracy** and **significantly lower inference latency than baselines across all seeds**.
>
> > **3. Comparison with newer speculative approaches**
>
> We have compared SpecSearch with **three recent speculative methods**: two advanced speculative decoding approaches—Lookahead [1] and Eagle [2]—and one speculative tree search method, Treebon [3]. As shown in Table 5 in 14604.pdf, SpecSearch achieves **1.74× to 2.73× speedups** over these baselines, highlighting its strong acceleration capability while preserving high reasoning quality.
>
> [1] Break the Sequential Dependency of LLM Inference Using Lookahead Decoding, ICML24.
>
> [2] EAGLE: Speculative Sampling Requires Rethinking Feature Uncertainty, ICML24
>
> [3] TreeBoN: Enhancing Inference-Time Alignment with Speculative Tree-Search and Best-of-N Sampling, 2024.10
>
> ### Weakness 3.
>
> > **4. A deeper comparison with SEED or other structured speculative decoding (SD) methods.**
>
> #### 1) The novelty over SEED and SD
> We discuss the novelty of SpecSearch compared to Scheduled Speculative Decoding (SEED) and existing SD methods, emphasizing key distinctions in our **bi-level speculative formulation**, **contextual verification**, **quality-preserving rejection strategies**, and **theoretical guarantees for reasoning quality**. Please refer to **Weakness 2** in **Response to Reviewer uX9o** for details.
>
> It is worth noting that the key distinction between SEED and standard SD methods lies in its use of **N small models** to **generate N token sequences in parallel at each node** of the reasoning tree, followed by a **Rounds-Scheduled strategy** to coordinate a shared large model for verification. However, in terms of the four aspects—**formulation**, **verification** and **rejection**, and **theoretical guarantees**—SEED remains consistent with SD methods.
>
> #### 2) The novelty over a recent structured speculative tree search method
> We discuss the novelty of SpecSearch compared to Treebon[3], emphasizing key distinctions in terms of **motivation**, **speculative formulation**, **rejection strategies**, and **theoretical guarantees**. Please refer to **Weakness 2** in **Response to Reviewer uX9o** for details.
>
> ### Weakness 4.
> > **5. Compare wall-clock time on real hardware with parallelization overhead considered**
>
> We measured wall-clock inference latency on a dual NVIDIA A800 GPU setup, including parallelization overhead. As shown in Table 6 in 14604.pdf, **SpecSearch achieves a 2.87$\times$ speedup in total latency**, fully accounting for parallelization costs.
>
> ### Weakness 5.
> > **6. Discussion on failure cases where the small model’s generation misleads the system**
>
> We discussed failure cases in Figure 10 in Appendix G.2 in our initial submission. Specifically, in this case, the small model misinterpreted the statement "A was born X years before B," reversing its meaning. Moreover, our **thought evaluator failed to identify this error** and **assigned a high score** of 0.89 to the incorrect reasoning step, ultimately leading to an incorrect final answer.
>
> ### Weakness 6.
> > **7. Speedup sensitivity to the small draft model’s performance**
>
> We have investigated SpecSearch’s performance using multiple small draft models. The results in Table 7 in 14604.pdf reveal that **SpecSearch achieves speedups ranging from 2.18$\times$ to 2.87$\times$**, underscoring its **robust acceleration capabilities across diverse small-model settings**.

---

### Official Review · Reviewer_uX9o · 2025-03-15

**Overall Recommendation:** 2

**Summary:**

This paper proposes SpecSearch to optimize thought generation through strategic collaboration between a small model and a large model at both thought and token levels. This approach efficiently produces high-quality reasoning thoughts. A key feature of SpecSearch is its quality-preserving rejection mechanism, which filters out low-quality thoughts, ensuring that only those meeting the standard of the large model are retained. Experimental results on the Qwen and Llama models show that SpecSearch achieves up to a 2.12× speedup compared to state-of-the-art methods, while maintaining comparable reasoning quality.

**Claims And Evidence:**

yes

**Essential References Not Discussed:**

no

**Experimental Designs Or Analyses:**

yes

**Methods And Evaluation Criteria:**

yes

**Other Comments Or Suggestions:**

See weaknesses part.

**Other Strengths And Weaknesses:**

Strengths:

The paper proposes a bi-level speculative thought generator that combines a small model and a large model at both thought and token levels. This design leverages the small model’s parallel generation efficiency for proposing diverse reasoning thoughts and the large model’s evaluation capability for filtering, achieving up to 2.12× speedup while maintaining comparable reasoning quality.


Weaknesses:

1. The experiments are limited to two mathematical reasoning datasets (only subsets of MATH and GSM8K) and structured tasks. The framework’s effectiveness on more complex reasoning tasks (e.g., AIME and Olympiad bench) remains unverified, raising questions about its effectiveness and generalization ability.


2. While the integration of speculative execution with tree-search-based reasoning is practical, the core idea of combining small and large models resembles existing speculative decoding techniques (e.g., token-level draft-then-verify). The claimed "first generalization" of speculative execution to reasoning lacks a clear distinction from prior work [1].

[1] Qiu J, Lu Y, Zeng Y, et al. Treebon: Enhancing inference-time alignment with speculative tree-search and best-of-n sampling[J]. arXiv preprint arXiv:2410.16033, 2024.

**Questions For Authors:**

See weaknesses part.

**Relation To Broader Scientific Literature:**

N/A

**Theoretical Claims:**

N/A

---

> ### Author Rebuttal · Authors · 2025-04-01
>
> # Response to Reviewer uX9o
>
> We sincerely thank the reviewer for the insightful and valuable feedback. We genuinely hope our response has addressed your concerns. If it has, we would be truly grateful if you would consider raising your score. If not, we warmly welcome any further suggestions and will continue doing our best to improve the submission.
>
> ### Weakness 1.
>
> > **1. The experiments are limited to only subsets of MATH and GSM8K. The framework’s effectiveness on more complex reasoning tasks (e.g., AIME and Olympiad bench) remains unverified.**
>
> We have conducted comprehensive evaluations across **three distinct dataset categories** to rigorously demonstrate the efficiency and generalizability of SpecSearch. Specifically, these include: (1) the **full GSM8K** dataset comprising 1,319 problems; (2) more challenging mathematical reasoning benchmarks, namely the **AIME** and **Olympiad** datasets; and (3) a **code-generation** benchmark. The results show that SpecSearch **consistently** and **significantly surpasses state-of-the-art approaches** across all three dataset categories, achieving speedups ranging from **2.04$\times$** to **2.84$\times$** while maintaining comparable reasoning accuracy. Due to limited space, please refer to **Weakness 1** in **Response to Reviewer cNpL** for detailed results.
>
> ### Weakness 2.
>
> > **2. The core idea of combining small and large models resembles existing speculative decoding (SD) techniques (e.g., token-level draft-then-verify).**
>
> We discuss the novelty of SpecSearch compared to existing SD techniques, emphasizing key distinctions in terms of **speculative formulation**, **verification** and **rejection strategies**, and **theoretical guarantees**. We will include the discussion in our revised paper.
>
> - **Bi-Level Speculative Formulation**: Unlike existing SD methods focused solely on tokens, SpecSearch treats both high-level thoughts and low-level tokens as bi-level speculative tasks. This enables (1) **Structural Alignment** with reasoning frameworks, where thoughts are fundamental units, and (2) **Compatibility** with standard SD methods through low-level token-level speculation.
> - **Contextual Verification for Higher Acceptance and Speedup**: Unlike existing SD methods that enforce strict token-level alignment, leading to frequent rejections, SpecSearch verifies the **contextual quality** of reasoning thoughts. This allows acceptance of correct but non-aligned outputs, substantially boosting acceptance rates and achieving significant speedups.
> - **Quality-Preserving Rejection Mechanism**: Unlike token-level rejection in standard SD methods, SpecSearch proposes **quality-preserving thought-level rejection** based on contextual quality. It discards entire thoughts only when their quality is lower than the large model’s, ensuring high-quality reasoning throughout decoding.
> - **Theoretical Guarantee of Reasoning Quality**: While standard SD methods preserve token-level distributions, SpecSearch guarantees that reasoning quality remains comparable with outputs from the large model.
>
> > **3. The claimed "first generalization" of speculative execution to reasoning lacks a clear distinction from Treebon [1].** ([1] Treebon: Enhancing inference-time alignment with speculative tree-search and best-of-n sampling.)
>
> We discuss the novelty of SpecSearch compared to Treebon [1], emphasizing key distinctions in terms of **motivation**, **speculative formulation**, **rejection strategies**, and **theoretical guarantees**. We will include the discussion in our revised paper.
>
> - **Distinct Motivation**: Unlike Treebon, which targets accelerating best-of-n sampling through speculative rejection combined with tree search, SpecSearch is the first to well generalize speculative execution to tree-based LLM reasoning.
> - **Bi-Level Speculative Formulation**: Treebon treats fixed-length token sequences as speculative tasks, while SpecSearch introduces a **flexible bi-level approach**—modeling full reasoning thoughts as high-level tasks and tokens as low-level ones. Unlike Treebon’s fixed-length design, SpecSearch leverages LLMs' reasoning capabilities to generate semantically coherent thoughts of dynamic length.
> - **Quality-Preserving Rejection Mechanism**: Treebon rejects a fixed proportion of token sequences using a preset threshold. SpecSearch, instead, scores reasoning thoughts and **adaptively rejects those with lower contextual quality based on the large model’s reasoning quality**, enabling finer control and better quality preservation.
> - **Theoretical Guarantee**: Unlike Treebon, which lacks theoretical guarantees, SpecSearch provides **formal assurance that reasoning quality remains uncompromised**, matching that of the large model's outputs.

---

### Official Review · Reviewer_cNpL · 2025-03-17

**Overall Recommendation:** 4

**Summary:**

This paper proposes a speculative search framework, that extends speculative decoding framework to reasoning chains. SpecSearch framework works by rejecting and selecting both at thought and token levels. The authors show both performance and speedup improvements.

**Claims And Evidence:**

The claim is that SpecSearch - applying speculative decoding at both thought and token level improves reasoning quality and provides speedup. It is backed by evidence across both Qwen and LLama models across both MATH-100 and GSM8K-100 datasets.

**Essential References Not Discussed:**

None that I am aware of

**Experimental Designs Or Analyses:**

The experimental designs and corresponding analyses are sound despite one significant concern. The authors only evaluate the approach on 100 randomly selected subsets of both MATH and GSM8K datasets. While the claims of latency can be adequately claimed, the accuracy claims might not be significant and might need more samples for a convincing argument.

**Methods And Evaluation Criteria:**

Yes

**Other Comments Or Suggestions:**

(Covered above)

**Other Strengths And Weaknesses:**

(Covered above)

**Questions For Authors:**

(Covered above)

**Relation To Broader Scientific Literature:**

The contributions of this paper can be broadly useful. Speculative decoding is a standard technique both in industry and academia, and its extension to reasoning chains, if claims hold, has wide implications.

**Theoretical Claims:**

I did not check the theoretical claims.

---

> ### Author Rebuttal · Authors · 2025-04-01
>
> # Response to Reviewer cNpL
>
> We sincerely thank the reviewer for the insightful, valuable, and positive comments. We address the concerns in detail as follows. We sincerely hope that our response could properly address your concerns. If so, we would deeply appreciate it if you could raise your score. If not, please let us know your further concerns, and we will continue actively responding to your comments and improving our submission.
>
> ### Weakness 1.
>
> > **1. The authors only evaluate the approach on subsets of both MATH and GSM8K datasets. The accuracy claims might need more samples for a convincing argument.**
>
> We have conducted comprehensive evaluations across **three distinct dataset categories** to rigorously demonstrate the efficiency and generalizability of SpecSearch. Specifically, these include: (1) the **full GSM8K** dataset comprising 1,319 problems; (2) more challenging mathematical reasoning benchmarks, namely the **AIME** and **Olympiad** datasets; and (3) a **code-generation** benchmark. As illustrated in the following Table, SpecSearch **consistently** and **significantly surpasses state-of-the-art approaches** across all three dataset categories, achieving speedups ranging from **2.04$\times$** to **2.84$\times$** while maintaining comparable reasoning accuracy. These findings highlight SpecSearch's versatility and robustness, demonstrating substantial improvements in inference speed with minimal or no compromise in accuracy **across diverse tasks**. We will include the results in our revised paper.
>
> - **Setup** Throughout our experiments, we utilize quantized versions of Qwen2.5-72B-Instruct and Qwen2.5-7B-Instruct as the large and small language models, respectively. Additionally, we incorporate MATH-psa as the Process Reward Model and employ beam search as the search algorithm.
> - **Results**
>     - **Full GSM8K Dataset (1,319 Problems)**: SpecSearch achieves a substantial **2.84$\times$ speedup** compared to the AR baseline, with only a minimal accuracy reduction of 0.83%. This result highlights SpecSearch’s capability to effectively scale to larger problem sets while preserving high reasoning accuracy.
>     - **High-Difficulty Mathematics (AIME and Olympiad Bench)**: We conduct experiments on the AIME and Olympiad Bench (OE_TO_maths-zh_CEE) datasets. Notably, SpecSearch **maintains identical accuracy** to the SpS method while achieving **speedups of 1.21$\times$ and 1.37$\times$**, respectively. These results demonstrate the method’s effectiveness in handling challenging, competition-level mathematics problems.
>     - **Code Generation (HumanEval)**: To assess SpecSearch beyond mathematical reasoning, we evaluate its performance on the HumanEval code-generation benchmark. The results show that SpecSearch achieves a **2.16$\times$ speedup** over the AR without any reduction in accuracy. Furthermore, it **surpasses the SpS by 1.22% in accuracy** while simultaneously delivering a **1.41$\times$ speedup**. These results underscore SpecSearch's strong generalization capabilities across diverse domains.
>
> | **MATH   Dataset** | **GSM8K-1319** |  |  |  |
> |:---:|:---:|:---:|:---:|:---:|
> | Methods | Reasoning      Accuracy (\%) | Average   Inference      Latency (s) | Speedup      (vs AR) | Speedup      (vs SpS) |
> |  |  |  |  |  |
> | AR | 96.66 | 144.63 | NA | 0.48 |
> | SpS | 96.66 | 70.04 | 2.06 | NA |
> | SpecSearch (Ours) | 95.83 | **50.99** | **2.84** | **1.37** |
> | **MATH Dataset** | **AIME** |  |  |  |
> | Methods | Reasoning      Accuracy (\%) | Average   Inference      Latency (s) | Speedup      (vs AR) | Speedup      (vs SpS) |
> |  |  |  |  |  |
> | AR | 16.67 | 562.89 | NA | 0.57 |
> | SpS | 13.33 | 318.71 | 1.77 | NA |
> | SpecSearch (Ours) | 13.33 | **264.44** | **2.13** | **1.21** |
> | **MATH Dataset** | **Olypamid Bench** |  |  |  |
> | Methods | Reasoning      Accuracy (\%) | Average   Inference      Latency (s) | Speedup      (vs AR) | Speedup      (vs SpS) |
> |  |  |  |  |  |
> | AR | 63.75 | 358.44 | NA | 0.67 |
> | SpS | 58.75 | 241.80 | 1.48 | NA |
> | SpecSearch (Ours) | 58.75 | **176.02** | **2.04** | **1.37** |
> | **Coding Dataset** | **HumanEval** |  |  |  |
> | Methods | Reasoning      Accuracy (\%) | Average   Inference      Latency (s) | Speedup      (vs AR) | Speedup      (vs SpS) |
> |  |  |  |  |  |
> | AR | 85.37 | 342.18 | NA | 0.65 |
> | SpS | 84.15 | 223.30 | 1.53 | NA |
> | SpecSearch (Ours) | 85.37 | **158.43** | **2.16** | **1.41** |

---

> > ### Comment · Reviewer_cNpL · 2025-04-05
> >
> > I think this paper definitely is improved with more experiments. I am recommending an accept but with the caveat of me not knowing the literature as best as an expert in this area.

---

> > > ### Author Response · Authors · 2025-04-05
> > >
> > > Dear Reviewer cNpL,
> > >
> > > Thank you very much for your positive feedback and for recommending our paper for acceptance! We sincerely appreciate your thoughtful comments, and we are especially grateful for your acknowledgment of the improvements brought by the additional experiments. We will include these additional results in the revised version, which we believe will further enhance the clarity and completeness of our work. Once again, thank you for your time and thoughtful review.

---

### Decision · Program_Chairs · 2025-05-01

**Decision:**

Accept (poster)

**Comment:**

This paper introduces **SpecSearch**, a speculative search framework that accelerates tree-search-based reasoning in large language models (LLMs) by leveraging a small model to draft and verify reasoning thoughts at both the thought and token levels.

Reviewers **cNpL**, **EXez**, and **zUgA** praised the work for its novelty, theoretical guarantees, and strong empirical results. Key strengths highlighted include the broad applicability of speculative decoding to reasoning tasks and the paper’s well-structured presentation. Reviewer **uX9o** raised concerns about limited generalization (initially tested on subsets of MATH/GSM8K) and the novelty of the approach compared to speculative decoding (SD) and Treebon. The authors addressed these concerns in their rebuttal, and no further questions remain.

Given the consistently positive feedback from the reviewers, as well as the work’s completeness and innovation, I recommend **acceptance** of this paper.